

# Endolithic Boring Enhance the Deep-sea Carbonate Lithification on the Southwest Indian Ridge

Hengchao Xu, Xiaotong Peng*, Shun Chen, Jiwei Li, Kaiwen Ta, Mengran Du

Deep-sea Science Division, Institute of Deep-Sea Science and Engineering, Chinese Academy of Science, Sanya 572000,
China

*Correspondence to*: Xiaotong Peng (xtpeng@idsse.ac.cn)

**Abstract.** Deep-sea carbonates represent an important type of sedimentary rock due to their effect on the composition of upper oceanic crust and their contribution to deep-sea geochemical cycles. However, the lithification of deep-sea carbonates at the seafloor has remained a mystery for many years. A large lithified carbonate area, characterized by thriving benthic

faunas and tremendous amount of endolithic borings, was discovered in 2008, blanketed on the seafloor of ultraslow spreading Southwest Indian Ridge (SWIR). Macrofaunal inhabitants including echinoids, polychaetes, gastropods as well as crustaceans, are abundant in the sample. The most readily apparent feature of the sample is the localized enhancement of density around the borings. The boring features of these carbonate rocks and factors that may enhance deep-sea carbonate lithification are reported. We suggest that active boring may trigger the dissolution of the original calcite and thus accelerate

deep-sea carbonate lithification on mid-ocean ridges. Our study reports an unfamiliar phenomenon of non-burial carbonate lithification and interested by the observation that it is often associated with boring feature. These carbonate rocks may provide a novel mechanism for deep-sea carbonate lithification at the deep-sea seafloor and also illuminate the geological and biological importance of deep-sea carbonate rocks on mid-ocean ridges.

## 1 Introduction

Carbonate rocks and sediments formed of various types have been discovered on various mid-ocean ridges through dredging or drilling (Thompson et al., 1968; De et al., 1985; Cooke et al., 2004). These carbonates, which are important elements of the upper oceanic crust, cover approximately half of the area of the entire ocean floor. As such they may influence the composition of oceanic crust and alter the geochemical balance between the total amounts of calcium, magnesium, and carbon in oceanic waters (Holligan and Robertson, 1996; Rae et al., 2011; Yu et al., 2014; Anderson et al.,

25  1976).

Most carbonate in the deep sea is biogenic in origin and may involve diagenetic products that originates from calcareous biogenic debris. Porosity loss with increasing age and burial depth is associated with the transformation of deep-sea calcareous ooze to chalk and subsequently to limestones (Flügel, 2004). Nevertheless, the processes involved in the formation of deep-sea carbonate rocks remains controversial. It has commonly been assumed that deep-sea carbonate



lithification is driven by various processes, including gravitational compaction and pressure dissolution, and reprecipitation that takes place during burial (Croizéet al., 2013). Local dissolution and reprecipitation of biogenic calcite or aragonite from foraminifera, nanofossils, and pteropod oozes may serve to transform the original sediments to chalk or limestone (Schlanger and Douglas, 1974). These explanations, however, cannot completely explain the facts that (i) the degree of burial is

commonly inconsistent with the known burial depth and paleontological age (Schmoker and Halley, 1982), and (ii) lithified carbonate rocks found on the seafloor commonly show no evidence of ever being buried (Thompson et al., 1968). Lithification of deep-sea carbonates has also been associated with the submarine breakdown of basalt or prolonged exposure to the chemical gradients at the sediment-water interface (Pimm et al., 1971; Bernoulli et al., 1978). Nevertheless, the processes responsible for such seafloor lithification remain open to debate.

Burrowing and boring organisms play a critical role in sediment evolution because it enhances interactions between sediments, the interstitial waters and overlying water by changing the geochemical gradients in the sediment, restructuring bacterial communities, and influencing the physical characteristics of the sediment (Lohrer et al., 2004; Meysman et al., 2006; Barsanti et al., 2011; Lalonde et al., 2010). Although organismic burrowing and boring has already been recognized as a factor that may influence $CaCO_3$ sediment profiles (Emerson and Bender, 1981; Aller, 1982; Emerson et al., 1985; Green

et al., 1992), and may promote carbonate dissolution in coastal sediments (Gerino et al., 1998), little is known about the relationship between boring and bioturbation in semi-lithified and lithified carbonate rocks in deep sea settings.

This study is based on non-burial chalk samples that were collected in 2008 near a newly discovered hydrothermal vent on Southwest Indian Ridge (SWIR) during the DY115-20 cruise of R/V Dayang Yihao, which was conducted by the China Ocean Mineral Resource R&D Association (COMRA) (Fig. 1). These carbonate deposits, which are associated with a

thriving benthic biota, are characterized by numerous endolithic borings. This research attempted to explore the mystery of the non-burial carbonate lithification in deep-sea and to highlight the interactions that take place between the bioturbation and lithification on mid-ocean ridge.

## 2. Geological Setting

The Indian Ocean includes an important but poorly understood part of the global meridional ocean. The Southwest

Indian Ridge (SWIR), which is the major boundary between the Antarctic Plate and the African Plate, characterized by its ultraslow and oblique expansion, is one of the slowest-spreading ridges (1.4-1.6 cm/yr) in the global ocean ridge system (Dick et al., 2003). Three main ridge sections of eastern part of the SWIR are divided by the Gallieni Transform Fault (GTF) and the Melville Transform Fault (MTF) (Cannat et al., 1999). In 2008, a large lithified carbonate area, approximately 15 km long and 150 km wide in water 2000 to 2500 m deep, was found on segment 26 of the SWIR near a newly discovered

hydrothermal field (Fig. 1). It has been well proved that thriving biogenic bloom through the Indian Ocean were bursting during the Latest Miocene–Early Pliocene by ODP sediment sequences (Arumugm et al., 2014; Gupta et al., 2004; Rai and Singh, 2001; Singh et al., 2012). The bloom lead to significantly higher carbonate mass accumulation rates than present day





between 9.0 to 3.5 Ma and promote the high quantities of carbonates deposit at the seafloor (Gupta et al., 2004; Dickens and Owen, 1999).

## 3. Material and Methods

### 3.1 Sampling

5    The deep-sea carbonate samples were collected in 2008 by TV-grabs bucket operated from the R/V Da Yang Yi Hao. The survey sites coved an area approximately 15 km long and 150 km wide. When carbonate samples were spread on the deck, benthic organisms were usually evident among the fractured rocks. Samples were subsampled after recovery, and then stored at -20 ℃ in plastic bags for mineralogical and geochemical analysis. Subsamples for molecular phylogenetic analysis, that were kept in dry ice frozen and transported to the laboratory, were described by Li et al. (2014).

### 3.2 Computerized X-ray tomography (CT)

Quantitative measurement of the significance of biological influence is difficult because the physico-chemical properties around boring walls are dynamic. Computerized X-ray tomography is a non-destructive method that has been used in to measure various rock properties (e.g., bulk density, porosity, macropore size) by determining the numerical value of the X-ray attenuation coefficient. For relatively homogeneous marine sediments, this coefficient is expressed as Hounsfield units (HU), which is correlated with sample density (Michaud et al., 2003). In this study, computerized X-ray tomography measurements were performed using a GE Light Speed VCT instrument that is located in the Shanghai 10th People Hospital, Tongji University. CT images were computerized by reconstruction of the distribution function of the linear attenuation coefficient, each with a 64-slice system with $64 \times 0.625$ mm detector banks and a z-axis coverage of 40 mm. The slice thickness is 2.5 mm and the accuracy of distance measurements in the x and y-planes is 0.2 mm. The instrument operates at 140 kV, with a 10 mA current, and 1.5 s exposure.

CT images were further characterized by *Image J*, which is a public domain Java-based image processing program. Gray values, which correlated with the attenuation values and HU, were extracted to make a comparing description of the density changes of the samples. 40 CT images were selected, and each gray value inverted using min = 0 and max = 255, regardless of the data values; that is, the theoretical integrated density value without the carbonate sample will be close to zero. The calibration function was used to calibrate whole images to a set of density standards before extracting. After all images selected were calibrated, the integrated density of the rock around the borings can be calculated from the gray values. For this study, the gray values of the 10 pixels (approximately 0.3 cm, compared to the diameters of borings are approximately 0.9 cm) around the boring holes were measured. Additionally, randomly selected areas around the borings were selected as control controls.

25



### 3. 3 X-ray diffraction (XRD)

Small pieces of the samples, which were freeze-dried under vacuum conditions to avoid oxidation during drying, were thoroughly ground using a pestle and mortar to produce a fine-grained, uniform powder. These powders were analysed using a D/max2550VB3+/PC X-ray diffractometer (Rigaku Corporation) at 40 kV and 30 mA, which is housed at the State Key

Laboratory of Marine Geology at Tongji University.

### 3. 4 Scanning electron microscope (SEM)

Small fragments of the dried samples were fixed onto aluminum stubs with two-way adherent tabs, and allowed to dry overnight. They were then sputter coated with gold for 2-3 minutes. All samples were examined with a Philips XL-30 scanning electron microscope that is equipped with an EDAX energy dispersive X-ray spectrometer and analytical software

at the State Key Laboratory of Marine Geology, Tongji University and Department of Earth and Atmospheric Sciences, University of Alberta. Energy dispersive X-ray spectroscopy (EDS) was mainly qualitative because of the irregular surface topography of the samples. The SEM was operated at 15 kV with a working distance of 10 mm to provide optimum imaging and minimize charging and sample damage. For the X-ray analysis, an accelerating voltage of 20 kV was used in order to obtain sufficient X-ray counts.

**3.5 Element and isotope analysis**

After fusion of 0.1 g of sample material with 3.6 g of dilithium tetraborate at 1050 ℃ for ca. 16 min, major elements were measured using X-ray fluorescence Shimadzu XRF-1800 spectrometer at 40 kV and 95 mA that is located at Shanghai University. The trace element and rare earth element (REE) compositions of the samples were determined by inductively coupled plasma-mass spectrometry (ICP-MS) using a Thermo VG-X7 mass spectrometer at the State Key Laboratory of

Marine Geology, Tongji University. The samples for these analyses were dissolved using a solution of $HNO_3$ + HF on a hot plate. The eluted sample was then diluted with 2% $HNO_3$. The precision of the sample duplicates, as well as of the repeated analyses, was better than 5%.

Stable oxygen and carbon isotope ratios of bulk samples were measured using a Finnigan MAT252 isotope ratio mass spectrometer equipped with a Kiel III carbonate device at the State Key Laboratory of Marine Geology, Tongji University.

Bulk samples were oven-dried at 60 ℃. Analytical precision was monitored using the Chinese national carbonate standard, GBW04405. Conversion of measurements to the Vienna Peedee Belemnite (PDB) scale was performed using NBS-19 and NBS-18.



## 4. Results

### 4.1 Macroscopic observations

The rocks retrieved from the SWIR are characterized by complex honeycombed structures and the ferromanganese crusts commonly encrust the surface and inner surface of carbonate rocks (Fig 2). Macrofaunal inhabitants, including echinoids, polychaetes, gastropods and crustaceans, which are usually recognized as successful boring classes in marine sediments (Kristensen and Kostka, 2013), are abundant in the carbonate sample with its density up to 12 per dm2 (Fig. 2). Boring holes drilled by benthic fanua, in straight, branched, or J- and U-shaped (Fig. 2e) with several millimeters to 2 cm in diameter, commonly penetrate 6 to 10 cm into the samples. The area that surround the hole is usually brighter than that away from the boring which may herald a different degree of lithification.

The most prominent feature of the carbonate is that thriving benthic biota are intimately related with the carbonate ever since the biogenic debris had deposited on the sea floor because three types of boring are easily identified. Borings with living organisms can be categorized explicitly as fresh borings (Fig 2c, d). The second type is considered to be the vacant boring which is filled by gray excrements (Fig 2b). Thin black ferromanganese crusts commonly encrust the surface of carbonate and the inner surface of empty borings (Fig. 2a, c, d) and this can be classified as the third type of boring. The gray excrements and (or) encrusted black ferromanganese crusts should be subsequently deposited after the boring formation, with the fact of which the latter two borings can be classified as old generations. Thus, the influence of bioturbation in this area is most likely a continuous process during the early lithification and could play a significant role in both geological and biological processes

### 4. 2 Enhanced lithification around boring

Computerized X-ray tomography (CT) was used for better characterization of local changes of density in the carbonate rocks, which would reflect their degree of lithification. A darker color in the tomographic cross-section image of the sample represents lower attenuation and thus lower density and higher porosity. The most readily apparent feature of the CT image is the localized enhancement of density around the boring (Fig. 3c, d). The shape of the area with a higher density around the hole is triangular, quadrangular, hexagonal, round or irregular (Fig. 4). The integrated density extracted from the tomographic cross-section images intuitively reveals the relative density change around the boring. The density enhancement can increase by 120% relative to the surrounding sediment (Fig. 5), which provides robust evidence for the significant influence of bioturbation in deep-sea carbonate lithification. In addition, the results of CT also show that the density is generally higher at the bottom than at the top of the carbonate rocks (Fig. 3a, b).

### 4. 3 Mineralogy

Based on XRD and element analyses, the sediment and rock samples consist almost entirely of calcite and detectable quart, halite which is typical for deep-sea chalk defined as "soft, pure, earthy, fine-textured, usually white to light gray or



buff limestone of marine origin, consisting almost wholly (90-99%) of calcite, formed mainly by accumulation of calcareous tests of floating micro-organisms (chiefly foraminifers) and of comminuted remains of calcareous algae (such as coccoliths and rhabdoliths) set in a structureless matrix of very finely crystalline calcite (Wolfe, 1968; Flügel, 2004)". Thin section and scanning electron photomicrographs show that biogenic components, mainly planktonic foraminifera (*Globigerina bulloides*)

and cocolithophorid (*Coccolithus pelagicus*) dominate (Fig 6). The presence of *Globigerina bulloides* indicates that the lithification history of carbonate rocks are less than 5 Ma (Pliocene-Recent) old. Therefore, carbonate deposit on the SWIR should be the bioclastic deposition from the productivity related events 'biogenic bloom' to large part of Indian Ocean during middle Miocene to the early Pliocene (Singh et al., 2012; Rai and Singh, 2001; Gupta et al., 2004; Arumugm et al., 2014).

Although it is difficult to separate and quantify the small tests from the very fine matrix, the carbonate exhibit a relatively high test to matrix ratio which is representative of deep-sea chalk (Fig. 6a). Original skeletal grains are held together by cement. Body chambers in the foraminiferal tests, for instance, are partially filled by calcite cements (Fig. 6b, c). Accretionary overgrowth of calcite around the foraminifera test also is common which are observed coating by coccoliths (Fig. 6c). Dissolution of the coccolith plates is evident both on the surface of the thin black ferromanganese crust and in the

interior of carbonate rocks (Fig 6e). The gray excrements filling the boring consist primarily of plates of coccolithophorids (*Calcidiscus leptoporus, Emiliania huxleyi* and *Gephurocapsa oceanica*). The smooth surfaces of plates in gray excrements reveal that dissolution of the original plates has already occurred (Fig 6f), which also presented dipartite evolutionary of diagenesis compared to the chalk.

### 4.4 Geochemistry and isotope analysis

Three portions of sample (chalk, gray excrements and thin black ferromanganese crust) exhibit similar elemental concentration patterns for high CaO content, reflecting the strong dilution of biogenic calcium in bulk samples. One of the main character of major and rare element is highly variable of Sr concentrations in different portion of carbonate. The storage of Sr on the seafloor is mainly account for the substitution for Ca in calcium carbonate while the diagenetic recrystallization, resulting in Sr loss from the sediment (Plank and Langmuir, 1998; Qing and Veizer, 1994). The loss of

Sr/Ca in chalk compared to the gray excrements could also a response to the lithification of carbonate. Although biogenic calcium diluted the detrital REE fraction, it made little direct contribution to bulk REE concentrations (Xiong et al., 2012). REE patterns of the three portions of sample do not exhibit any hydrothermal anomalies, e.g. positive Eu anomaly, but inherit the characteristics of sea water by enrichment of HREE compared with LREE and negative Ce anomaly (except the ferromanganese crust) (Fig. 7). The influence of nearby hydrothermal system and other detrital input to the study carbonate

area should negligible during the lithification history.

The $\delta^{13}C_{PDB}$ values of 46 bulk samples are -0.37 to 1.86‰ which are typical for biogenic carbonates. These samples have a relatively narrow $\delta^{18}O_{PDB}$ range of 1.35 to 3.79‰. There is an evident depletion of carbon and oxygen isotopic values of gray excrements compared to chalk which reflects the bioleaching effect by benthic fauna. The bulk $\delta^{13}C_{PDB}$ values of



chalk and gray excrements are positively correlated with bulk $\delta^{18}O_{PDB}$ values (r = 0.91) (Fig. 8), which reveals minor environmental influence on early lithification and endolithic boring should be a critical factor influence the lithification.

## 5. Discussion

### 5.1 Endolithic boring in carbonate rock on SWIR

Boring holes generally with several millimeters to 2 cm in diameter, commonly penetrate 6 to 10 cm into the chalks and ultimately reach a density up to 12 per $dm^2$ (Fig. 2a). Although the real depth of each boring hole is difficult to determine accurately because the full lengths of boring usually are not apparent, the volume occupied by the boring can be estimated in a simple model. The boring in straight, branched, or J- and U-shaped (Fig 2e) are simplified to a cylinder with the diameter of 1 cm and height of 6 cm, which are the median value of the boring holes. In this model, 1 $dm^2$ surface area which can harbor 12 boring holes on the surface may reach to 0.226 $dm^3$ boring space. An important conclusion can be deduced from this model that the carbonate substratum were reconstructed by the boring to a great extent.

Several boring purposes are served for the benthic animals such as gaseous exchange, food transport, gamete transport, transport of environmental stimuli, and removal of metabolites (Kristensen and Kostka, 2013). Carbonate mass accumulation during the Latest Miocene–Early Pliocene at Indo-Pacific provided the bioclastic deposition to the sea floor (Singh et al., 2012; Rai and Singh, 2001; Gupta et al., 2004; Arumugm et al., 2014). These carbonate sediments deposit on the seafloor thus form a favorable environment for benthic fauna (fig 2). Polychaetes, taking the most successful boring class for example (Díaz-Castañeda and Reish, 2009), are abundant and conventionally produce J- or U- shaped boring extended as long as several decimeters in carbonate sample (Fig. 2 c, e). During the frequent construction and maintenance of boring, not only the carbonate reworking and bio-mixing occurs by boring, the redox and geochemical parameters, including pH, were also assumed to oscillate around the bioturbate structure (Furukawa, 2001). Relic borings allow sea water to directly penetrate into carbonate rocks, which is benefit to the precipitation of black ferromanganese crusts on the inner surface of boring (Fig. 2a, c). Moreover, boring organisms excrete mucus to garden their boring holes by incorporating organic matter into the walls (Dworschak et al., 2006; Koller et al., 2006), the mucus layers lined on the inner side of boring walls usually are as thick as 5 μm and are composed of protein-rich mucopolysaccharide (Petrash et al., 2011). The mucus layer may friendly help constructing a favorable site for the accumulation of metallic ions through organo-metallic complexation or chelation at suitable Eh, pH and redox conditions (Lalonde et al., 2010; Banerjee, 2000).

In addition to the boring activity, benthic fauna ingest and excrete the substrate which usually serve the boring holes as traps for fecal pellets (Fig 2b) (Hydes, 1982; Aller and Aller, 1986). Although benthic fauna ingest organically enriched particles, thus removing the organic matter, the bulk sample is often still enriched in residual fecal material (Dauwe et al., 1998). Regardless, organic matter influenced by bioturbation and delivered as biodeposits in surface sediments, and vice versa, may therefore create a dynamic and heterogeneous chemical, physical and biological micro-environment in the deep-sea carbonate area. Eventually, a microenvironment friendly for heterotrophic microorganism may be formed in the



carbonate due to the redistribution of organic particles (Li et al., 2014). Alternatively, bacteria and organic detritus are considered to the major source of benthic fanua in deep-sea (Raghukumar et al., 2001). Thus, a balanced ecological sustainability are established by the carbonate deposits and the continuous biological process which may be largely influenced the lithification history of the carbonate.

## 5. 2 The roles of endolithic borings in lithification of carbonate rocks on the SWIR

Abundant endolithic borings, as well as a benthic fauna present on a cross section of a carbonate rock (e.g. polychaete worm, Fig. 2c), the dissolution of coccolith plates observed by SEM (Fig 6), and enhancement of density around boring holes commonly observed from CT images (Fig 3, 4), as well as elemental composition change between different portion of carbonate (Table 1) provided robust evidence for the significant role of bioturbation in present lithified deep-sea carbonates. Water depth of the carbonate rock area on SWIR varies approximately from 2000 to 2500 meters (Fig 1), which is above the calcite saturation horizon (Broecker et al., 1982). In this range of water depth, the key point of carbonate lithification is how the original tests or plates are dissolved under saturation condition. Although less stable $CaCO_3$ phase (e.g., biogenic, high-Mg calcites) may dissolve above the calcite saturation horizon (Jahnke and Jahnke, 2004), they are not likely to happen here since our samples are low-Mg calcites. As a general rule, compaction takes place with the gradually increasing of overburden pressure and resulting in losing of porosity through mechanical and chemical compaction in moderate-deep burial stage. However, present carbonate samples shows they have never been buried. The lithification of carbonate may be different from carbonate from other deep sea carbonate. Elements and isotope results reveal that minor external impact on the early lithification. Thus, the simultaneously activities of both thrive benthic fanua and lithification of carbonate should unavoidable have some connections with the fact that the construction and ventilation of boring by benthic fauna could fundamentally alter biogeochemical processes and produce lateral heterogeneity intensify the redistribution of pore water fluids and could create ecological niches for microbial life (Ghirardelli, 2002; Koretsky et al., 2013).

Providing that biogenically-produced $CaCO_3$ particles typically have very large surface areas due to the presence of pores in foraminiferal tests and coccolithophorid plates which may be increasingly exposed to pore waters as the primary particles are broken and dissolution proceeds (Jahnke and Jahnke, 2004). The organic matter should have been significantly low in the deep-sea sediment. Endolithic borings, as discussed above, reconstructed the carbonate substratum to a great extent and resulted in a fundamentally alteration of sedimentary environment. The aerobic respiration of bioturbated organic particles like mucus and fecal pellets would positively contribute to the aerobic respiration of bioturbated organic particles by heterotrophic (micro)organisms (Lohrer et al., 2004), whose reaction product $CO_2$ may be responsible for the lower pH porewater around the boring hole relative to the inner carbonate sediment, which may drive the dissolution of original calcite in microenvironment (Emerson and Bender, 1981; Aller, 1982; Kristensen, 2000). Thus, active bioturbation in carbonate rocks may provide a feasible pathway for the dissolution of tests or plates at such depth. The local elevated concentration of dissolved $CO_2$ in pore water would trigger the dissolution of the original $CaCO_3$ phases, which is consistent with the results



of SEM observation. For instance, Body chambers in the foraminiferal tests are partially filled by calcite cements (Fig. 6c, Fig. 7c), which is believed to be derived internally through solution transfer (Durney, 1972).

In addition, relic borings make sea water directly penetrate into carbonate rocks and lead to the precipitation of black ferromanganese crusts on the inner surface of boring hole. The microbial oxidation of $Fe^{2+}$ and $Mn^{2+}$ in these sites would

also greatly accelerate the dissolution of $CaCO_3$ fossils (Emerson and Bender, 1981; Aller and Rude, 1988). Furthermore, thin ferromanganese crust and grey sediments prevent the rapid ion exchange between bottom water and pore water within carbonate rocks, which may cause $Ca^{2+}$ and $CO_3^{2-}$, the production of $CaCO_3$ dissolution, diffuse toward to the interior of carbonate rocks, and lead to the reprecipitation of calcites as cements in carbonate rocks.

## 6. Conclusions

A lithified carbonate area characterized by active endolithic boring was investigated and discussed regarding the area's biological and geological interactions. Although the effect of different parameters influenced by boring cannot easily be differentiated in study of natural samples, available evidences show that active endolithic borings may trigger the dissolution of the original calcite above the saturation horizon and thus enhance deep-sea carbonate lithification on mid-ocean ridges. The novel mechanism proposed here for non-burial carbonate lithification at the deep-sea seafloor sheds light on the

potential interactions between deep-sea biota and sedimentary rocks, and also illuminate the geological and biological importance of deep-sea carbonate rocks on mid-ocean ridges.

## Acknowledgments

Special thanks go to all the participants of the cruise of R/V DayangYihao conducted by China Ocean Mineral Resource R&D Association (COMRA) in 2008. The authors also would like to acknowledge Dr. Sui Wan at Tongji

University for his help with the calcareous fossil analysis. Financial support for this research came from "Strategic Priority Research Program" of the Chinese Academy of Science (Grant No. XDB06020201), "National Key Basic Research Program of China" (2015CB755905), "Natural Science Foundation of Hainan Province, China" (20164175). We are indebted to Prof. Brian Jones at University of Alberta for his valuable suggests on the manuscript.

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

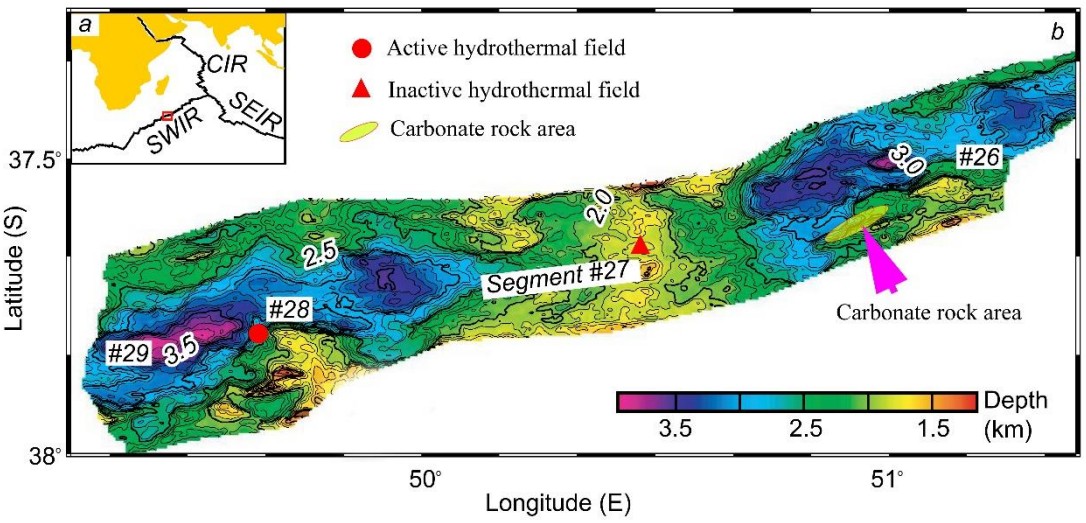

.

**Figure 1: (a) Location of study area on the Southwest Indian Ridge. (b) Bathymetric map of area which show the location of the**
**carbonate rock area (green ellipse), the active hydrothermal field (red triangle), and the inactive hydrothermal field (red cycle).**



**Figure 2: Deep-sea carbonate rocks collected from the SWIR. (a) A carbonate rock sample shows empty boring holes are partly covered by ferromanganese crusts. (b) Straight and branched borings are infilled by grey sediments. (c) Abundant borings, as well as a benthic fauna (polychaete worm), are present on a cross section of a carbonate rock. (d) An echinoid, together with other benthic faunas, borings on a carbonate rock with the honeycombed structures and encrusted by thin ferromanganese crusts. (e)**



**Sketch for different endolithic boring structures in deep-sea carbonate rocks collected from the SWIR. Scale bar of a, c are 5cm, and the b, d is 3 cm.**

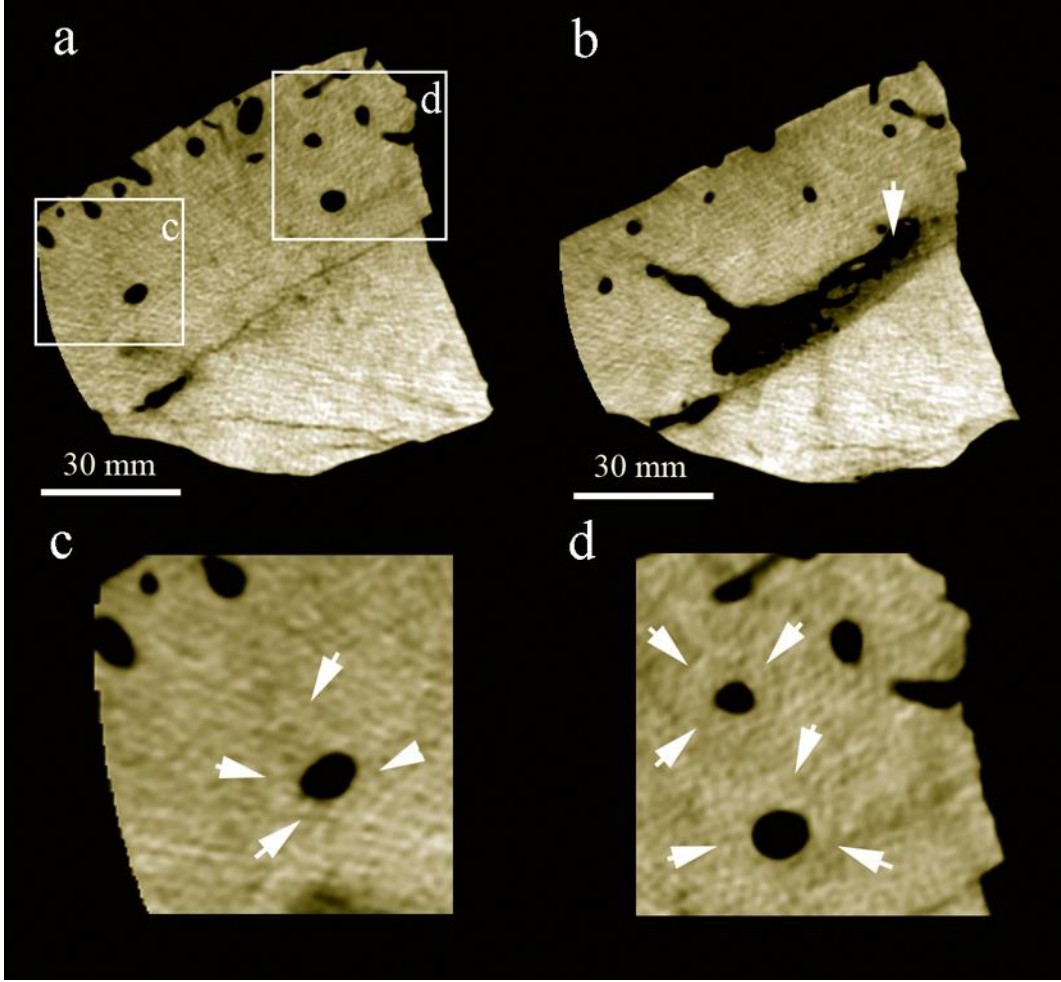

**Figure 3: (a, b) Tomographic cross-sections of a sample show the density is higher at the bottom than at the top. Darker colors represent lower attenuation and thus lower density and higher porosity. Boring holes are black in the figure. (c, d) The enlargement of Fig.6A shows the elevated density around the boring holes.**




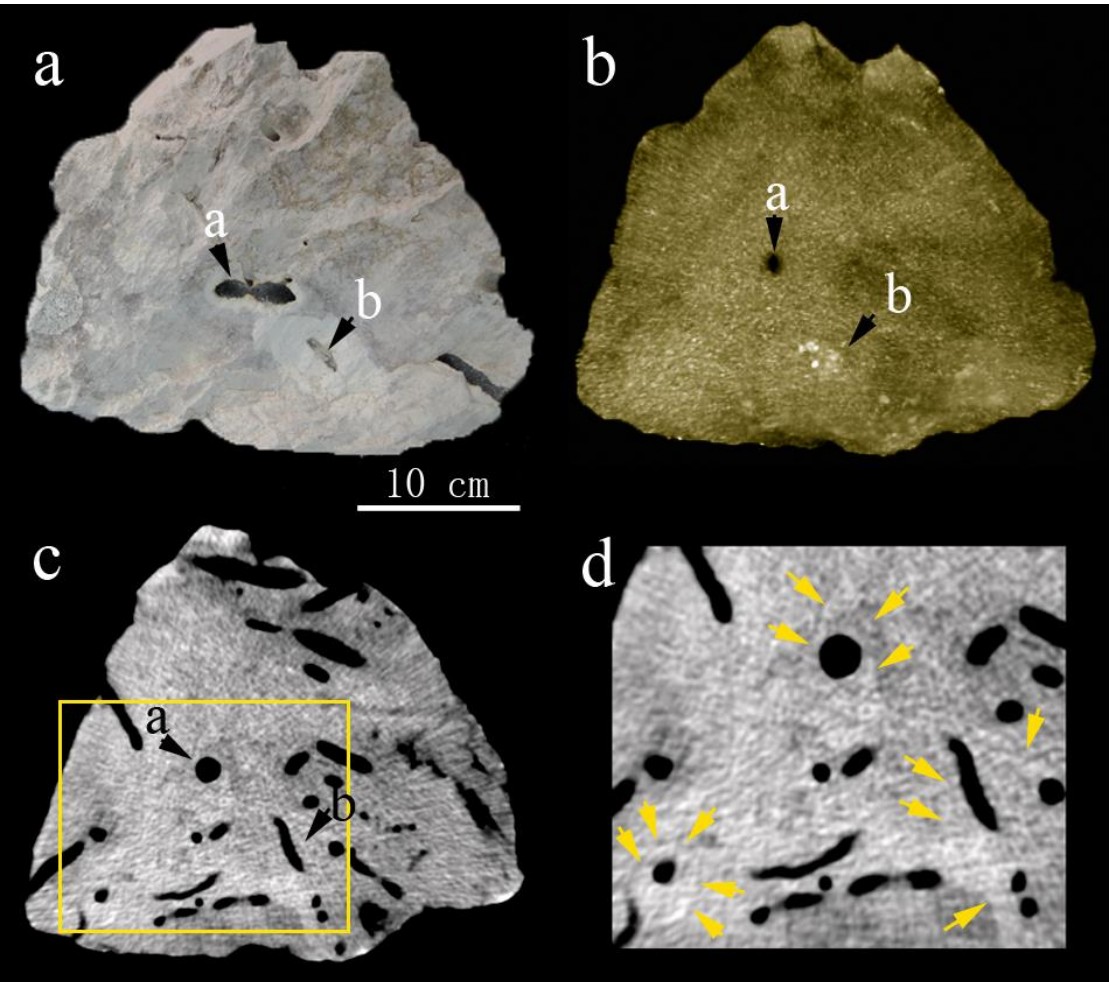

**Figure 4: (a) A hand specimen shows the enhancement of brightness associated with burrow structures. (b) 3D reconstruction of the sample shows enhancement of density around the burrows. Arrows point out the location of burrows. Darker colors represent lower attenuation and thus lower density and higher porosity. Burrows are black in the figure. (c) Tomographic cross-section of the sample reveals that abundant burrows are clearly present in the interior of the samples. Higher density areas with triangular, hexagonal and irregular shapes are visible around the burrows. (d) The enlargement of Fig. 4c.**





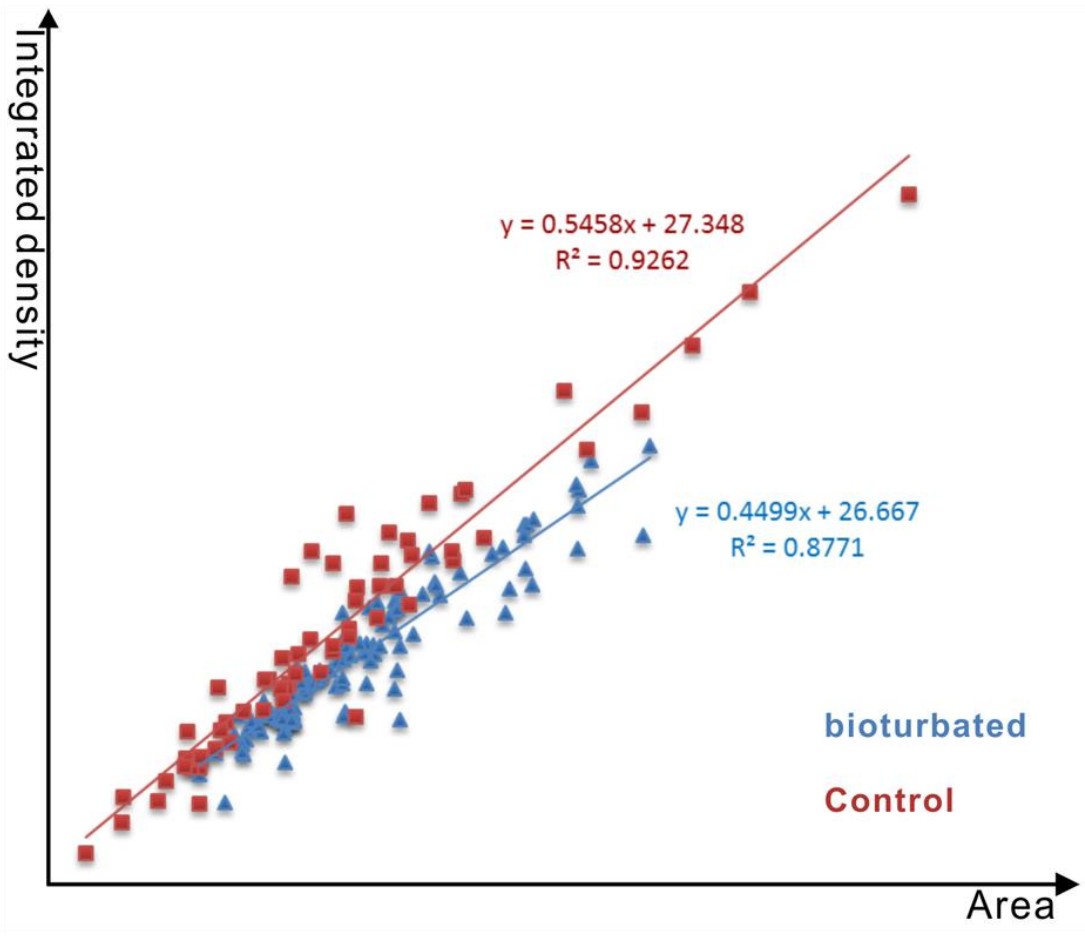

**Figure 5: Statistical analysis of integrated density extracted from selected area around boring holes and paralleled undisturbed area clearly shows enhancement of density around the holes. Both images were inverted so that bigger slope means a darker color in original CT image. The total number of analyzed holes=113.**





**Figure 6: (a) photomicrograph of thin sections of carbonate rocks shows a relatively high test (arrows) to matrix (m) ratio. (b) Scanning electron micrograph reveals abundant micritic carbonate particles (arrows) with many plates of coccoliths in the interior of carbonate rocks. (c) Scanning electron micrograph shows overgrowths of calcites on the foraminiferal in the interior of carbonate rocks. (d) Scanning electron micrograph shows dissolution of coccoliths in the interior of carbonate rocks. (e) Scanning electron micrograph shows the surface of carbonate rock covered by thin ferromanganese crusts (c). Arrow points out the**





dissolution of the coccoliths. (f) Scanning electron micrograph shows grey sediments which infill the boring. Smooth surfaces of the coccoliths indicate that the dissolution commonly occurs.

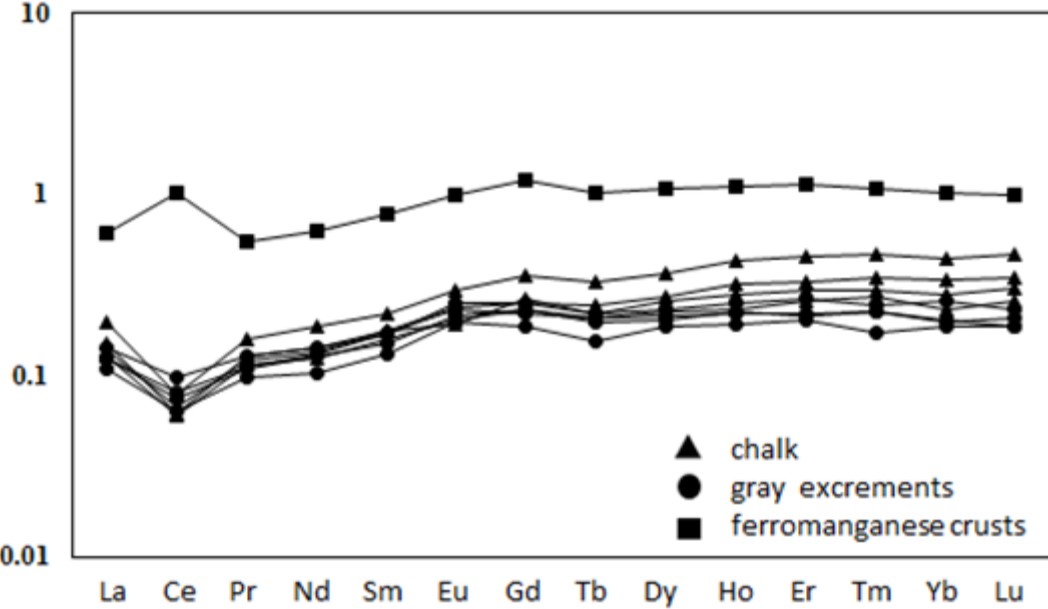

Figure 7: PAAS-normalized REE distribution patterns of selected samples from the SWIR.





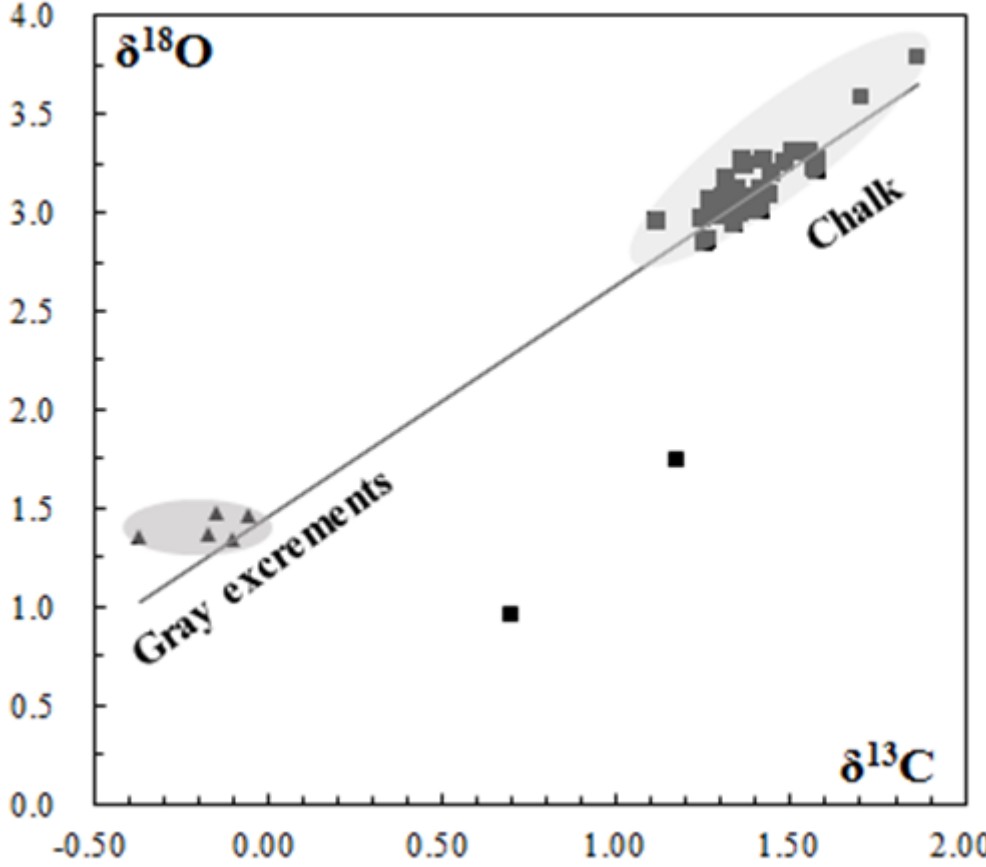

**Figure 8 Oxygen and carbon isotopic composition of carbonate samples from the SWIR. The gray excrements contain the light carbon and oxygen isotopic values compared to the chalk. The bulk $\delta^{13}C_{PDB}$ values of chalk and gray excrements are positively correlated with bulk $\delta^{18}O_{PDB}$ values (r=0.91).**



| Sample NO. | CaO | SiO$_2$ | Al$_2$O$_3$ | Na$_2$O | SO$_3$ | Fe$_2$O$_3$ | MnO | K$_2$O | P$_2$O$_5$ | Cr$_2$O$_3$ | L.O.I | Total |
|---|---|---|---|---|---|---|---|---|---|---|---|---|
| C1 | 87.60 | 5.01 | 1.60 | 1.26 | 1.01 | 0.99 | 0.08 | 0.14 | 0.12 | 0.07 | 2.10 | 100.00 |
| C2 | 89.39 | 4.73 | 1.54 | 0.49 | 1.10 | 0.91 | 0.07 | 0.09 | 0.12 | 0.06 | 1.50 | 100.00 |
| C3 | 87.81 | 5.02 | 1.64 | 1.01 | 0.91 | 1.08 | 0.09 | 0.13 | 0.14 | 0.07 | 2.10 | 100.00 |
| C4 | 88.35 | 4.46 | 1.53 | 1.25 | 1.06 | 0.88 | 0.06 | 0.15 | 0.11 | 0.07 | 2.07 | 100.00 |
| C5 | 90.70 | 4.09 | 1.04 | 0.45 | 0.84 | 1.09 | N.D. | 0.12 | 0.08 | 0.09 | 1.50 | 100.00 |
| C6 | 90.91 | 4.42 | 1.12 | 0.44 | 0.61 | 1.11 | 0.05 | 0.16 | 0.11 | 0.09 | 0.99 | 100.00 |
| C7 | 90.49 | 4.37 | 1.14 | 0.32 | 1.21 | 1.18 | 0.08 | N.D. | 0.10 | 0.07 | 1.04 | 100.00 |
| C8 | 89.33 | 4.88 | 1.18 | 0.40 | 1.37 | 1.24 | 0.06 | 0.15 | 0.09 | 0.06 | 1.23 | 100.00 |
| M1 | 84.31 | 4.64 | 1.22 | 0.50 | 2.18 | 3.12 | 1.57 | 0.14 | 0.20 | N.D. | 1.96 | 99.83 |

| Sample NO. | Li | Be | Sc | V | Cr | Co | Ni | Cu | Zn | Rb | Sr | Y | Zr | Nb |
|---|---|---|---|---|---|---|---|---|---|---|---|---|---|---|
| C1 | 12.394 | 0.151 | 1.327 | 9.067 | 6.101 | 7.499 | 16.927 | 16.284 | 8.453 | 2.272 | 1415.280 | 7.398 | 5.971 | 0.774 |
| C2 | 14.326 | 0.030 | 1.541 | 9.584 | 6.823 | 8.314 | 14.787 | 22.730 | 12.296 | 1.971 | 1570.712 | 7.794 | 6.433 | 0.710 |
| C3 | 9.924 | 0.060 | 1.452 | 10.544 | 7.200 | 16.402 | 13.308 | 14.540 | 6.429 | 2.073 | 1408.932 | 8.692 | 6.449 | 0.841 |
| C4 | 5.187 | 0.070 | 1.229 | 8.446 | 5.837 | 5.697 | 10.295 | 13.203 | 8.925 | 3.008 | 1355.301 | 6.697 | 4.917 | 0.650 |
| C5 | 9.269 | 0.050 | 1.427 | 6.355 | 4.520 | 3.033 | 9.568 | 27.786 | 10.845 | 2.794 | 1105.468 | 12.841 | 6.884 | 0.549 |
| C6 | 7.282 | 0.150 | 1.307 | 6.643 | 4.289 | 3.621 | 8.299 | 15.830 | 9.795 | 3.092 | 1071.291 | 9.646 | 6.613 | 0.579 |
| C7 | 6.185 | 0.150 | 1.556 | 7.123 | 4.938 | 6.265 | 23.184 | 12.959 | 15.942 | 2.823 | 1071.415 | 11.043 | 6.335 | 0.579 |



| Sample NO. | | | | | | | | | | | | | |
|---|---|---|---|---|---|---|---|---|---|---|---|---|---|
| C8 | 8.970 | 0.160 | 1.804 | 8.559 | 5.833 | 8.389 | 22.501 | 16.538 | 11.005 | 3.067 | 1210.755 | 17.159 | 8.369 | 0.842 |
| M1 | 17.389 | 0.270 | 2.805 | 76.871 | 7.806 | 432.418 | 111.399 | 29.926 | 38.920 | 4.272 | 1389.487 | 32.831 | 21.581 | 3.863 |

| Sample NO. | Mo | In | Cs | Ba | La | Ce | Pr | Nd | Sm | Eu | Gd | Tb | Dy | Ho | Er |
|---|---|---|---|---|---|---|---|---|---|---|---|---|---|---|---|
| C1 | 0.342 | 0.010 | 0.101 | 161.531 | 4.724 | 5.981 | 0.965 | 4.342 | 0.824 | 0.231 | 1.076 | 0.151 | 0.955 | 0.221 | 0.613 |
| C2 | 0.340 | 0.000 | 0.110 | 157.972 | 4.802 | 6.363 | 0.990 | 4.482 | 0.950 | 0.250 | 1.020 | 0.160 | 1.010 | 0.220 | 0.620 |
| C3 | 0.391 | 0.010 | 0.100 | 171.936 | 5.397 | 7.801 | 1.122 | 4.867 | 0.921 | 0.270 | 1.182 | 0.170 | 1.081 | 0.250 | 0.751 |
| C4 | 0.220 | 0.010 | 0.180 | 131.332 | 4.188 | 4.987 | 0.870 | 3.528 | 0.730 | 0.210 | 0.880 | 0.120 | 0.880 | 0.190 | 0.570 |
| C5 | 0.150 | 0.010 | 0.180 | 14.327 | 5.757 | 4.879 | 1.137 | 4.879 | 0.978 | 0.259 | 1.167 | 0.190 | 1.267 | 0.319 | 0.938 |
| C6 | 0.160 | 0.010 | 0.180 | 13.665 | 4.818 | 4.768 | 1.017 | 4.269 | 0.888 | 0.219 | 1.077 | 0.160 | 1.057 | 0.229 | 0.728 |
| C7 | 0.319 | 0.010 | 0.140 | 19.573 | 5.357 | 5.367 | 1.067 | 4.599 | 0.978 | 0.209 | 1.227 | 0.170 | 1.197 | 0.279 | 0.838 |
| C8 | 0.261 | 0.010 | 0.110 | 18.141 | 7.497 | 6.344 | 1.413 | 6.254 | 1.213 | 0.321 | 1.674 | 0.251 | 1.724 | 0.421 | 1.293 |
| M1 | 6.498 | 0.020 | 0.150 | 69.355 | 23.288 | 82.052 | 4.831 | 21.082 | 4.342 | 1.058 | 5.500 | 0.789 | 4.991 | 1.098 | 3.204 |

| Sample NO. | Tm | Yb | Lu | Hf | Ta | Tl | Pb | Th | U | ΣREE | LREE | HREE | LREE/HREE | δEu | δCe | Ce$_{anom}$ |
|---|---|---|---|---|---|---|---|---|---|---|---|---|---|---|---|---|
| C1 | 0.090 | 0.573 | 0.080 | 0.181 | 0.161 | 0.020 | 3.699 | 0.684 | 0.251 | 20.83 | 17.07 | 3.76 | 4.54 | 0.75 | 0.63 | -0.20 |
| C2 | 0.090 | 0.560 | 0.090 | 0.220 | 0.180 | 0.050 | 4.142 | 0.730 | 0.260 | 21.61 | 17.84 | 3.77 | 4.73 | 0.77 | 0.65 | -0.18 |
| C3 | 0.100 | 0.721 | 0.100 | 0.230 | 0.140 | 0.020 | 2.283 | 0.801 | 0.240 | 24.73 | 20.38 | 4.36 | 4.68 | 0.79 | 0.71 | -0.15 |
| C4 | 0.070 | 0.530 | 0.080 | 0.170 | 0.130 | 0.020 | 0.300 | 0.590 | 0.210 | 17.83 | 14.51 | 3.32 | 4.37 | 0.80 | 0.59 | -0.23 |
| C5 | 0.140 | 0.948 | 0.150 | 0.210 | 0.110 | 0.010 | 0.708 | 0.529 | 0.249 | 23.01 | 17.89 | 5.12 | 3.50 | 0.74 | 0.43 | -0.37 |
| C6 | 0.110 | 0.648 | 0.110 | 0.190 | 0.140 | 0.010 | 0.608 | 0.549 | 0.269 | 20.10 | 15.98 | 4.12 | 3.88 | 0.69 | 0.48 | -0.31 |





| | | | | | | | | | | | | | | | |
|---|---|---|---|---|---|---|---|---|---|---|---|---|---|---|---|
| **C7** | 0.120 | 0.778 | 0.130 | 0.209 | 0.100 | 0.010 | 1.905 | 0.549 | 0.259 | 22.32 | 17.58 | 4.74 | 3.71 | 0.58 | 0.50 | -0.30 |
| **C8** | 0.190 | 1.243 | 0.200 | 0.281 | 0.120 | 0.020 | 2.486 | 0.712 | 0.271 | 30.04 | 23.04 | 7.00 | 3.29 | 0.69 | 0.43 | -0.37 |
| **M1** | 0.429 | 2.865 | 0.429 | 0.669 | 0.180 | 0.200 | 60.850 | 4.003 | 0.858 | 155.96 | 136.65 | 19.31 | 7.08 | 0.66 | 1.74 | 0.24 |

$\delta Eu = (Eu)_N / 0.5(Sm + Nd)_N$

$\delta Ce = (Ce)_N / 0.5(La + Pr)_N$

$Ce_{anom} = \log(Ce/Ce^*)_{SN} = \log[2Ce_{SN} / (La_{SN} + Pr_{SN})]$

**Table 1 Geochemical character parameters of Carbonate rocks. C1-C4 represent the gray sediment infilled in the burrows, C5-C8 represent the white carbonate and M1 represent the thin black ferromanganese crusts, some white part may be mix unavoidable.**