# Peer review of "Endolithic Boring Enhance the Deep-sea Carbonate Lithification on the Southwest Indian Ridge"

_Biogeosciences, 2018_

## Referee Comment (RC1) · Anonymous Referee #1 · 26 Mar 2018

General comments:

authors: Xu et al.

This manuscript evaluates the impact of macrofaunal activity on the lithification of deep-sea carbonates. This appears to be an interesting and previously not very thoroughly explored topic, and has the potential to advance our understanding of both the impact of bioturbation, as well as the controls on carbonate lithification. I especially appreciate the fact that this paper is very to the point, and not overly extended (although it does require some in depth discussion on some points, see below). One immediate problem with the manuscript in its current form is the sometimes confusing writing

style. Additionally, most of the discussion rests on speculation, rather than data, which undermines the value of this manuscript. I believe the authors should expand their observations, and especially include a thorough, quantitative assessment to elucidate the relative importance of bioturbation for carbon lithification.

Major specific comments:

- Language: While this is not the case for the manuscript as a whole, there are a lot of parts which suffer from bad spelling and grammar (see the technical corrections for specific examples, that list is not exhaustive). Before this manuscript can be accepted, it should be thoroughly revised for language.

- Interpretation of the figures: I have some difficulties with following the interpretation of the images presented in Fig. 2 and 3. On P7L6-7 you state that it is difficult to know the real depth of each burrow. How exactly do you then go from the pictures in Fig.2a-d to the burrow shapes in Fig. 2e? How do you know they are J-shaped, and not just U-shaped, but broken off? Did you do this by eye, or did you use a cast, or scanning techniques? Please expand The same remark for Fig. 3, you state (P5L22-23) that 'the most readily observable feature is the localized enhancement of density around the boring'. I have looked at the images, and even with the arrows, I have a lot of difficulties finding these enhancements. Since most of your discussion rests on these observations, I think you should expand and more clearly explain on what this statement is based. You might want to consider playing with the contrast, or other visual techniques, to make these features more clear (cause now I cannot see them). Additionally, you need to expand on the statistics used to generate the linear correlations in Fig. 5.

- Discussion: I believe the biggest problem with this manuscript is the lack of data, and a quantitative discussion. I am supportive of the authors efforts, and I do believe that bioturbation could play a role in carbonate lithification. However, to make this case based on a few images, without any quantitative discussion, seems a bit of a shortcut. The authors should expand on these observations, and give a better mechanistic explanation on how bioturbation would enhance lithification, and include a thorough quantitative assessment of the relative importance of this process.

Minor specific comments:

- The authors use 'boring' throughout the manuscript when they discuss the burrowing of macrofaunal organisms. This is rather confusing, as in my experience, it is more common to use 'burrowing' and 'burrow(s)'. I would suggest to change this throughout the manuscript to improve readability

- Additionally, I had to google the word 'endolith'. While it is correctly used, I again would suggest to avoid the use of this word, and use the more common 'macrofaunal', 'benthic fauna' or others, to improve readability.

- Results: at the start of the results (P5L3) it is immediately stated that the macrofaunal burrows were lined with ferromanganese crusts. While this assumption is used aplenty throughout the manuscript (see for example Fig. 7), you do not provide evidence that these are indeed ferromanganese crusts. Please justify this assumption (or say it is just an assumption, but then explain why).

- P6L21-25: a lot of discussion about Sr, but it is not shown?

- Table 1 is too long to be readable. I would suggest to make this a supplementary table, and take out the most important trends and plot those in a figure.

- P6L33: how exactly does bioleaching deplete the isotopic value? P7L15-16: why do these carbonate deposits form a favourable environment? What is special about them?

- P7L5-11: I cannot follow the reasoning behind this estimation. You cannot find the real depth of the burrows, but assume 6 cm, with is the median value. How do you get a median value if you cannot determine the real burrow depth? How do you get to 12 holes per 1dm$^2$ surface? Should you not compare volume to volume? Please be more explicit.

Technical corrections:

- P1L11: 'macrofaunal inhabitants' -> not correct, better: 'benthic macrofauna'

- P1L15-16: 'Our study reports an unfamiliar phenomenon . . . and interested by the . . .' -> this sentence is very vague, and also wrong (what is interested?), please rephrase

- P1L16-17: 'These carbonate rocks may . . .' -> it is not the carbonate rocks that provide a mechanism, please rephrase. . .

- P1L29: 'remains' -> remain

- P2L10: 'Burrowing and boring' -> I believe these are synonyms 'because it enhances' -> because they enhance

- P2L13: 'organismic burrowing and boring' -> same remark as above, and organismic can be removed

- P21L21: 'between the bioturbation' -> 'between bioturbation'

- P2L30: 'it has been well proved' -> it has been well proven 'bursting' -> what does this sentence mean? Biogenic bloom was bursting?

- P2L32-P3L2: I understand the sentence, but he is not constructed correctly . . .

- P5L9: 'herald' -> indicate

- P5L10-11. What does this sentence mean?

- P5L14-16: the message you are trying to convey is unclear, please rephrase

- P5L31 'quart' -> quartz?

- P6L13-15: sentence does not make much sense

- P6L17: 'dipartite evolutionary of diagenesis' -> what does this mean?

- P6L21-25: 'character' -> characteristic, 'is highly variable of Sr' -> is the highly variable

Sr, 'different portion of' -> different portions of, 'mainly accounted for the substitution' -> mainly caused by substitution 'recrystallization, resulting in' -> recrystallization results in 'The loss of' -> the decrease of 'could also a response' -> could also be a response

- P6L31: statement needs a reference ('typical values for biogenic carbonates')

- P7L12 'several boring purposes are served for the benthic animals' -> does not make any sense, benthic fauna form burrows for certain purposes.

- P8L1-2: 'Alternatively, bacteria and organic detritus are considered to the major source of benthic fauna in deep-sea' -> this sentence means that benthic fauna originates from bacteria and organic detritus. While this is possibly true from an evolutionary perspective, I do not think this is what you want to say here . . .

---

## Referee Comment (RC2) · Anonymous Referee #2 · 30 Apr 2018

The manuscript titled "Endolithic Boring Enhance the Deep-sea Carbonate Lithification on the Southwest Indian Ridge" details observations and analyses of deep-sea carbonate samples that appear to be experiencing enhanced lithification associated with benthic faunal burrowing. The study employs computed X-ray tomography, visual and microscope observation, and geochemistry to evaluate the relationships between burrowing and the degree of carbonate lithification. The main conclusion is that burrowing is likely an important process accelerating carbonate lithification in the deep-sea.

The findings are intriguing and certainly of interest to a wide readership. My main reservations about this manuscript are twofold: (1) it is not immediately clear in some

CT-scan images that there are density contrasts (enhanced lithification) surrounding the burrows (Figure 4), and (2) it is not clear based on the data treatment that there is true statistical significance in the difference in density between bioturbated zones and control zones (Figure 5). See specific comments on these below.

If the authors can address the above major points then I can see this manuscript being of interest to a wide readership. I agree with the authors that burrowing-enhanced lithification would appear to be an important process if it can extrapolated to deep-sea carbonates world-wide.

While the English is already commendable for authors for whom English might be a second language, and it is possible to follow what the authors are saying throughout the manuscript, there remain minor issues with English throughout the manuscript. This should be easily fixed with a careful proofread by a native speaker.

I would consider the revisions required to address the general comments above and specific comments below to be major - significant blocks of text should be revised and additional statistical treatment should be applied to the dataset.

Specific Comments

Abstract, line 9: I'm not sure that one can say that lithification of deep-sea carbonates is a "mystery"; there is a respectable body of literature on lithification mechanisms and rates dating back over three decades. Perhaps better would be something like "the role of deep-sea macrofauna in their lithification remain poorly understood".

Abstract, line 12: "in the sample" makes it read as if there was only a single hand sample, when it appears that grab buckets provided multiple samples. This occurs elsewhere in the manuscript as well.

Abstract line 16: "interested by" doesn't make much sense - please re-phrase.

Abstract, last sentence: these results don't really speak to the importance of deep-sea carbonate sediments, simply the mechanisms of their formation. Please re-phrase.

Main text in general: while I find that the text is written in a clear and straightforwards manner, there remain minor grammatical errors throughout. If english is the authors' second language, then they should be commended - this manuscript already reads decently well. Nonetheless, further editing by a native english speaker is necessary to wrap up the grammatical loose ends that are apparent throughout the manuscript.

Page 2, line 24: it might offend researchers in diverse fields to say that the entire Indian Ocean is "poorly understood".

Page 2, line 30-34: grammatical issues, please re-phrase.

Materials and Methods: certain phrases in the methods have been reproduced word-for-word from previous work. For example, page 4, lines 12 through 14 - these identical lines are also found in Li et al. (2014). Even if the same methodology was used for both studies, it would be prudent to re-word the text in the methods.

Page 4, line 21: there should be no "elution" step in this technique. Also line 22, how was precision evaluated? Repeat measurements of standards? Finally, how were these measurements standardized - using multi-element solutions or by measurement of geostandards? The methods are not sufficiently detailed here.

From page 5 onwards: these are not ferromangense crusts in the strict sense of the word. Perhaps "Mn- and Fe-oxide precipitates" is a better term.

Page 5, line 10: I suggest re-phrasing this sentence.

Page 6, line 16-17: I suggest re-phrasing.

Page 6, line 24–25: you can't lose a ratio (but you can lower it).

Page 7 line 1: I suggest re-phrasing.

Discussion in general: it would be nice if the authors could elaborate on why a decrease in carbonate saturation state (leading to dissolution) promotes lithification (as opposed to an increase in carbonate saturation state leading to precipitation). Also,

while aerobic respiration decreases the local carbonate saturation state, sulfate reduction will increase it. Can the authors include a statement about oxygen penetration and the depth of sulfate reduction (even if it is simply based on the findings of others in similar settings)?

Figure 1 Legend: The legend indicates that the red triangle is an inactive hydrothermal field while the caption indicates that it is active - this contradiction needs to be resolved. Also at the end it should read "red circle".

Figure 2e should have a scale bar.

Figure 4b: Contrary to the caption, it is difficult to see any enhanced of density in this image.

Figure 4c and d: what do the different arrows represent? In a related vein, for Figures 4 b, c, and d in general - the areas of higher density are not obvious at all. Perhaps circle them or find some better way of highlighting these areas? Also could another presentation method be employed (e.g., an additional panel with contrast adjustments to better show the differences, perhaps shown alongside an un-modified version of the same figure for traceability)?

Figure 5: This is not a statistical analysis in the sense that it does not provide any measure of confidence in the comparison between the two slopes (e.g. whether they can be considered different with 95% confidence). For this you would need to use something like the function "polyfit" in MATLAB (for example). No statistical evidence is presented that these slopes are indeed different... this is a major point as the paper hinges on the importance of burrowing effects.

Figure 8: As a Kiel carbonate device was used, these are not "bulk" C isotope measurements, but $C_{carb}$ measurements (same for oxygen isotopes). That is to say, organic matter in the sample is not measured during the analyses when a Kiel carbonate device is used, only carbonate - this should be clarified.

[Figure]

---

## Author Comment (AC1) · 27 May 2018

RC2: The manuscript titled "Endolithic Boring Enhance the Deep-sea Carbonate Lithification on the Southwest Indian Ridge" details observations and analyses of deep-sea carbonate samples that appear to be experiencing enhanced lithification associated with benthic faunal burrowing. The study employs computed X-ray tomography, visual and microscope observation, and geochemistry to evaluate the relationships between burrowing and the degree of carbonate lithification. The main conclusion is that burrowing is likely an important process accelerating carbonate lithification in the deep-sea. The findings are intriguing and certainly of interest to a wide readership.

[Figure]

Reply: We are very thankful to the anonymous reviewer for constructive feedbacks and insightful comments on our manuscript.

RC2: My main reservations about this manuscript are twofold: (1) it is not immediately clear in some CT-scan images that there are density contrasts (enhanced lithification) surrounding the burrows (Figure 4).

Reply: Thanks for your reminding. In order to make the density contrasts clearly, we pick the pixel values of the CT image to contrast the change of density around burrow. It is showed by the line scan profiles that pixel values around the bioturbated area is higher than matrix indicating the localized enhancement of density around burrows (Fig 3d). 3D reconstruction of the sample by CT analysis also has been added in revised manuscript (Fig 4). You can find new figures in response to specific comments.

RC2: (2) it is not clear based on the data treatment that there is true statistical significance in the difference in density between bioturbated zones and control zones (Figure 5). See specific comments on these below. If the authors can address the above major points then I can see this manuscript being of interest to a wide readership. I agree with the authors that burrowing-enhanced lithification would appear to be an important process if it can extrapolated to deep-sea carbonates world-wide.

Reply: Thanks very much for your advisable suggestions to promote the quality of data treatment. In revised paper, function "polyfit" in MATLAB as you advised is used to generate the polynomial p(area) that is a best fit for the data for integrated density (with 95% confidence bounds). The functions of bioturbated zones and control zones are discrete with a statistical significance (Fig 5). We show an example of the density change around the burrow by line scanning in Fig 3d. When comes to the Fig 5, whose data are generated from 113 burrows, the statistical results support our conclusion that macrofaunal burrowing enhance the deep-sea carbonate lithification on the Southwest Indian Ridge.

RC2: While the English is already commendable for authors for whom English might be

a second language, and it is possible to follow what the authors are saying throughout the manuscript, there remain minor issues with English throughout the manuscript. This should be easily fixed with a careful proofread by a native speaker.I would consider the revisions required to address the general comments above and specific comments below to be major - significant blocks of text should be revised and additional statistical treatment should be applied to the dataset.

Reply: Thank you very much for your reminding in English language. We took the utmost care to refine our English in the revised version.

Specific Comments RC2: Abstract, line 9: I'm not sure that one can say that lithification of deep-sea carbonates is a "mystery"; there is a respectable body of literature on lithification mechanisms and rates dating back over three decades. Perhaps better would be something like "the role of deep-sea macrofauna in their lithification remain poorly understood".

Reply: The sentence has been revised. the role of deep-sea macrofauna in carbonate lithification remains poorly understood.

RC2: Abstract, line 12: "in the sample" makes it read as if there was only a single hand sample, when it appears that grab buckets provided multiple samples. This occurs elsewhere in the manuscript as well.

Reply: Thanks for reminding. It has been modified and we have checked the whole manuscript to avoid this mistake.

RC2: Abstract line 16: "interested by" doesn't make much sense - please re-phrase.

Reply: The sentence has been deleted.

RC2: Abstract, last sentence: these results don't really speak to the importance of deep-sea carbonate sediments, simply the mechanisms of their formation. Please re-phrase.

Reply: The sentence has been revised. Macrofaunal burrowing provides a novel driving force for deep-sea carbonate lithification at the seafloor, illuminating the geological and biological importance of deep-sea carbonate rocks on global mid-ocean ridges.

RC2: Main text in general: while I find that the text is written in a clear and straight-forwards manner, there remain minor grammatical errors throughout. If english is the authors' second language, then they should be commended - this manuscript already reads decently well. Nonetheless, further editing by a native english speaker is necessary to wrap up the grammatical loose ends that are apparent throughout the manuscript.

Reply: whole manuscript has been deeply checked for English language.

RC2: Page 2, line 24: it might offend researchers in diverse fields to say that the entire Indian Ocean is "poorly understood".

Reply: Thanks for the comment. This sentence has been deleted.

RC2: Page 2, line 30-34: grammatical issues, please re-phrase.

Reply: It has been modified. This phenomenon known as "biogenic bloom" promoted significantly high quantities of carbonates deposit at the seafloor between 9 to 3.5Ma (Gupta et al., 2004; Dickens and Owen, 1999)

RC2: Materials and Methods: certain phrases in the methods have been reproduced wordfor-word from previous work. For example, page 4, lines 12 through 14 - these identical lines are also found in Li et al. (2014). Even if the same methodology was used for both studies, it would be prudent to re-word the text in the methods.

Reply: The text has been re-phrased. Small fragments of the dried samples were fixed onto aluminum stubs with two-way adherent tabs, and allowed to dry overnight. They were sputter coated with gold for 2-3 minutes before being examined on a Philips XL-30 scanning electron microscope equipped with an accelerating voltage of 15kV at the State Key Laboratory of Marine Geology, Tongji University.

RC2: Page 4, line 21: there should be no "elution" step in this technique. Also line 22, how was precision evaluated? Repeat measurements of standards? Finally, how were these measurements standardized - using multi-element solutions or by measurement of geostandards? The methods are not sufficiently detailed here.

Reply: Sorry for my mistake. There is no "elution" step.

Analytical precision was monitored using the Chinese national carbonate standard, GBW04405. Conversion of measurements to the Vienna Peedee Belemnite (PDB) scale was performed using NBS-19 and NBS-18.

RC2: From page 5 onwards: these are not ferromangense crusts in the strict sense of the word. Perhaps "Mn- and Fe-oxide precipitates" is a better term.

Reply: thanks for your advice. It has been changed.

RC2: Page 5, line 10: I suggest re-phrasing this sentence.

Reply: This sentence has been re-phrased. Burrows can be classified in three categories.

RC2: Page 6, line 16-17: I suggest re-phrasing.

Reply: This sentence has been re-phrased. Smooth surfaces of the coccoliths in gray excrements reveal that dissolution commonly occurs influenced by bioleaching of benthic fauna (Fig 6f)

RC2: Page 6, line 24–25: you can't lose a ratio (but you can lower it).

Reply: Thanks for reminding. It has been corrected.

RC2: Page 7 line 1: I suggest re-phrasing.

Reply: This sentence has been re-phrased. Positive correlation of $\delta13C$PDB and$\delta18O$PDB values of chalk and gray excrements (r = 0.91) reveals minor environmental influence on early lithification (Fig. 8) and bioturbation should be a critical factor

during the lithification.

RC2: Discussion in general: it would be nice if the authors could elaborate on why a decrease in carbonate saturation state (leading to dissolution) promotes lithification (as opposed to an increase in carbonate saturation state leading to precipitation).

Reply: Thanks for your comments. We have made efforts to explain the dissolution and reprecipitation of calcite to cement in revised manuscript. The dissolution of carbonate in the ocean is primarily controlled by the degree of pore water undersaturation with respect to the biogenic carbonate phase. Bioturbation could redistribute the organic matter around the burrow. Thus, oxidation of organic matter will accelerate the concentration of pore water $CO_2$ leading to the undersaturation of calcites. Furthermore, thin Mn- and Fe oxide precipitates may prevent the rapid ion exchange between bottom water and pore water within carbonate rocks because larger grain surfaces and porosity of fine-grained poorly sorted carbonate oozes compared to Mn- and Fe oxide precipitates. The products of $CaCO_3$ dissolution may trend to diffuse toward to the interior of carbonate rocks, and lead to an enhanced $CO_3^{2-}$ ion gradient in pore water profile and ultimately promoting the reprecipitation of calcites as cements around the burrows in carbonate rocks.

RC2: Also, while aerobic respiration decreases the local carbonate saturation state, sulfate reduction will increase it. Can the authors include a statement about oxygen penetration and the depth of sulfate reduction (even if it is simply based on the findings of others in similar settings)?

Reply: Thanks for your valuable comment. We cannot exclude the potential that sulfate reduction had happened in our chalk samples. This can be illustrated for the present study by examining the observed variations in ion content of the pore water. However, carbonate samples here were collected by TV-grabs bucket. It is too difficult to take the measurement of pore water chemistry. Several literatures support our discussion that pore-water $CO_2$ by oxidation of organic matter is responsible for the carbonate

dissolution. (Broecker and Peng 1982; Jahnke et al. 1994; Noé et al. 2006; Croizé et al. 2013). Metabolic activity may disintegration of organic material causing dissolution of carbonate and increasing the degree of supersaturation. In the condition that bioturbation processes succeed in redistribution of organic matter around the burrow, concentration of $CO_2$ in pore water could increase. Although we could not elaborate the influence of sulfate reduction, aerobic respiration is reasonable to the decrease of carbonate saturation state.

RC2: Figure 1 Legend: The legend indicates that the red triangle is an inactive hydrothermal field while the caption indicates that it is active - this contradiction needs to be resolved. Also at the end it should read "red circle".

Reply: It has been corrected. The red circle is active hydrothermal field and the red triangle indicates inactive fields.

RC2: Figure 2e should have a scale bar.

Reply: The scale bar has been added. Scale bar of Figure 2e is 3cm. We make the estimation of burrow depths from the CT images which are usually of 6- 10cm penetrating into chalk.

RC2: Figure 4b: Contrary to the caption, it is difficult to see any enhanced of density in this image. Figure 4c and d: what do the different arrows represent? In a related vein, for Figures 4 b, c, and d in general - the areas of higher density are not obvious at all. Perhaps circle them or find some better way of highlighting these areas? Also could another presentation method be employed (e.g., an additional panel with contrast adjustments to better show the differences, perhaps shown alongside an un-modified version of the same figure for traceability)? Reply: Thanks for your advice. Both Fig. 3 and Fig. 4 have been changed. We pick the pixel values of the CT image to contrast the changes of density around burrows. It is showed by the line scan profiles that pixel values around the bioturbated area is higher than matrix indicating the localized enhancement of density around burrows.

Figure 5: This is not a statistical analysis in the sense that it does not provide any measure of confidence in the comparison between the two slopes (e.g. whether they can be considered different with 95% confidence). For this you would need to use something like the function "polyfit" in MATLAB (for example). No statistical evidence is presented that these slopes are indeed different... this is a major point as the paper hinges on the importance of burrowing effects.

Reply: Thanks very much for your valuable suggestions. The figure has been revised. Difference can be showed with the function "polyfit" with 95% confidence intervals.

RC2: Figure 8: As a Kiel carbonate device was used, these are not "bulk" C isotope measurements, but C_carb measurements (same for oxygen isotopes). That is to say, organic matter in the sample is not measured during the analyses when a Kiel carbonate device is used, only carbonate - this should be clarified.

Reply: Sorry for the mistake. It has been corrected.

Please also note the supplement to this comment:
https://www.biogeosciences-discuss.net/bg-2018-46/bg-2018-46-AC1-supplement.pdf
* * *
[Figure]

Fig. 1.

[Figure]

**Fig. 2.**

[Figure]

a

b

10 cm

c

d

Fig. 3.

[Figure]

**Fig. 4.**

[Figure]

**Fig. 5.**

Fig. 6.

**Fig. 7.**

[Figure]

**Fig. 8.**

[Figure]

[Figure]

**Fig. 9.**

[Figure]

**Fig. 10.**

**Supplement:**

| Sample NO. | CaO | SiO$_2$ | Al$_2$O$_3$ | Na$_2$O | SO$_3$ | Fe$_2$O$_3$ | MnO | K$_2$O | P$_2$O$_5$ | Cr$_2$O$_3$ | L.O.I | Total |
|---|---|---|---|---|---|---|---|---|---|---|---|---|
| C1 | 87.60 | 5.01 | 1.60 | 1.26 | 1.01 | 0.99 | 0.08 | 0.14 | 0.12 | 0.07 | 2.10 | 100.00 |
| C2 | 89.39 | 4.73 | 1.54 | 0.49 | 1.10 | 0.91 | 0.07 | 0.09 | 0.12 | 0.06 | 1.50 | 100.00 |
| C3 | 87.81 | 5.02 | 1.64 | 1.01 | 0.91 | 1.08 | 0.09 | 0.13 | 0.14 | 0.07 | 2.10 | 100.00 |
| C4 | 88.35 | 4.46 | 1.53 | 1.25 | 1.06 | 0.88 | 0.06 | 0.15 | 0.11 | 0.07 | 2.07 | 100.00 |
| C5 | 90.70 | 4.09 | 1.04 | 0.45 | 0.84 | 1.09 | N.D. | 0.12 | 0.08 | 0.09 | 1.50 | 100.00 |
| C6 | 90.91 | 4.42 | 1.12 | 0.44 | 0.61 | 1.11 | 0.05 | 0.16 | 0.11 | 0.09 | 0.99 | 100.00 |
| C7 | 90.49 | 4.37 | 1.14 | 0.32 | 1.21 | 1.18 | 0.08 | N.D. | 0.10 | 0.07 | 1.04 | 100.00 |
| C8 | 89.33 | 4.88 | 1.18 | 0.40 | 1.37 | 1.24 | 0.06 | 0.15 | 0.09 | 0.06 | 1.23 | 100.00 |
| M1 | 84.31 | 4.64 | 1.22 | 0.50 | 2.18 | 3.12 | 1.57 | 0.14 | 0.20 | N.D. | 1.96 | 99.83 |

| Sample NO. | Li | Be | Sc | V | Cr | Co | Ni | Cu | Zn | Rb | Sr | Y | Zr | Nb |
|---|---|---|---|---|---|---|---|---|---|---|---|---|---|---|
| C1 | 12.394 | 0.151 | 1.327 | 9.067 | 6.101 | 7.499 | 16.927 | 16.284 | 8.453 | 2.272 | 1415.280 | 7.398 | 5.971 | 0.774 |
| C2 | 14.326 | 0.030 | 1.541 | 9.584 | 6.823 | 8.314 | 14.787 | 22.730 | 12.296 | 1.971 | 1570.712 | 7.794 | 6.433 | 0.710 |
| C3 | 9.924 | 0.060 | 1.452 | 10.544 | 7.200 | 16.402 | 13.308 | 14.540 | 6.429 | 2.073 | 1408.932 | 8.692 | 6.449 | 0.841 |
| C4 | 5.187 | 0.070 | 1.229 | 8.446 | 5.837 | 5.697 | 10.295 | 13.203 | 8.925 | 3.008 | 1355.301 | 6.697 | 4.917 | 0.650 |
| C5 | 9.269 | 0.050 | 1.427 | 6.355 | 4.520 | 3.033 | 9.568 | 27.786 | 10.845 | 2.794 | 1105.468 | 12.841 | 6.884 | 0.549 |
| C6 | 7.282 | 0.150 | 1.307 | 6.643 | 4.289 | 3.621 | 8.299 | 15.830 | 9.795 | 3.092 | 1071.291 | 9.646 | 6.613 | 0.579 |
| C7 | 6.185 | 0.150 | 1.556 | 7.123 | 4.938 | 6.265 | 23.184 | 12.959 | 15.942 | 2.823 | 1071.415 | 11.043 | 6.335 | 0.579 |

| | | | | | | | | | | | | | |
|---|---|---|---|---|---|---|---|---|---|---|---|---|---|
| **C8** | 8.970 | 0.160 | 1.804 | 8.559 | 5.833 | 8.389 | 22.501 | 16.538 | 11.005 | 3.067 | 1210.755 | 17.159 | 8.369 | 0.842 |
| **M1** | 17.389 | 0.270 | 2.805 | 76.871 | 7.806 | 432.418 | 111.399 | 29.926 | 38.920 | 4.272 | 1389.487 | 32.831 | 21.581 | 3.863 |

| Sample NO. | Mo | In | Cs | Ba | La | Ce | Pr | Nd | Sm | Eu | Gd | Tb | Dy | Ho | Er |
|---|---|---|---|---|---|---|---|---|---|---|---|---|---|---|---|
| **C1** | 0.342 | 0.010 | 0.101 | 161.531 | 4.724 | 5.981 | 0.965 | 4.342 | 0.824 | 0.231 | 1.076 | 0.151 | 0.955 | 0.221 | 0.613 |
| **C2** | 0.340 | 0.000 | 0.110 | 157.972 | 4.802 | 6.363 | 0.990 | 4.482 | 0.950 | 0.250 | 1.020 | 0.160 | 1.010 | 0.220 | 0.620 |
| **C3** | 0.391 | 0.010 | 0.100 | 171.936 | 5.397 | 7.801 | 1.122 | 4.867 | 0.921 | 0.270 | 1.182 | 0.170 | 1.081 | 0.250 | 0.751 |
| **C4** | 0.220 | 0.010 | 0.180 | 131.332 | 4.188 | 4.987 | 0.870 | 3.528 | 0.730 | 0.210 | 0.880 | 0.120 | 0.880 | 0.190 | 0.570 |
| **C5** | 0.150 | 0.010 | 0.180 | 14.327 | 5.757 | 4.879 | 1.137 | 4.879 | 0.978 | 0.259 | 1.167 | 0.190 | 1.267 | 0.319 | 0.938 |
| **C6** | 0.160 | 0.010 | 0.180 | 13.665 | 4.818 | 4.768 | 1.017 | 4.269 | 0.888 | 0.219 | 1.077 | 0.160 | 1.057 | 0.229 | 0.728 |
| **C7** | 0.319 | 0.010 | 0.140 | 19.573 | 5.357 | 5.367 | 1.067 | 4.599 | 0.978 | 0.209 | 1.227 | 0.170 | 1.197 | 0.279 | 0.838 |
| **C8** | 0.261 | 0.010 | 0.110 | 18.141 | 7.497 | 6.344 | 1.413 | 6.254 | 1.213 | 0.321 | 1.674 | 0.251 | 1.724 | 0.421 | 1.293 |
| **M1** | 6.498 | 0.020 | 0.150 | 69.355 | 23.288 | 82.052 | 4.831 | 21.082 | 4.342 | 1.058 | 5.500 | 0.789 | 4.991 | 1.098 | 3.204 |

| Sample NO. | Tm | Yb | Lu | Hf | Ta | Tl | Pb | Th | U | $\Sigma$REE | LREE | HREE | LREE/HREE | $\delta$Eu | $\delta$Ce | Ce$_{anom}$ |
|---|---|---|---|---|---|---|---|---|---|---|---|---|---|---|---|---|
| **C1** | 0.090 | 0.573 | 0.080 | 0.181 | 0.161 | 0.020 | 3.699 | 0.684 | 0.251 | 20.83 | 17.07 | 3.76 | 4.54 | 0.75 | 0.63 | -0.20 |
| **C2** | 0.090 | 0.560 | 0.090 | 0.220 | 0.180 | 0.050 | 4.142 | 0.730 | 0.260 | 21.61 | 17.84 | 3.77 | 4.73 | 0.77 | 0.65 | -0.18 |
| **C3** | 0.100 | 0.721 | 0.100 | 0.230 | 0.140 | 0.020 | 2.283 | 0.801 | 0.240 | 24.73 | 20.38 | 4.36 | 4.68 | 0.79 | 0.71 | -0.15 |
| **C4** | 0.070 | 0.530 | 0.080 | 0.170 | 0.130 | 0.020 | 0.300 | 0.590 | 0.210 | 17.83 | 14.51 | 3.32 | 4.37 | 0.80 | 0.59 | -0.23 |
| **C5** | 0.140 | 0.948 | 0.150 | 0.210 | 0.110 | 0.010 | 0.708 | 0.529 | 0.249 | 23.01 | 17.89 | 5.12 | 3.50 | 0.74 | 0.43 | -0.37 |
| **C6** | 0.110 | 0.648 | 0.110 | 0.190 | 0.140 | 0.010 | 0.608 | 0.549 | 0.269 | 20.10 | 15.98 | 4.12 | 3.88 | 0.69 | 0.48 | -0.31 |

| | | | | | | | | | | | | | | | | |
|---|---|---|---|---|---|---|---|---|---|---|---|---|---|---|---|---|
| **C7** | 0.120 | 0.778 | 0.130 | 0.209 | 0.100 | 0.010 | 1.905 | 0.549 | 0.259 | 22.32 | 17.58 | 4.74 | 3.71 | 0.58 | 0.50 | -0.30 |
| **C8** | 0.190 | 1.243 | 0.200 | 0.281 | 0.120 | 0.020 | 2.486 | 0.712 | 0.271 | 30.04 | 23.04 | 7.00 | 3.29 | 0.69 | 0.43 | -0.37 |
| **M1** | 0.429 | 2.865 | 0.429 | 0.669 | 0.180 | 0.200 | 60.850 | 4.003 | 0.858 | 155.96 | 136.65 | 19.31 | 7.08 | 0.66 | 1.74 | 0.24 |

$\delta Eu = (Eu)_N / 0.5(Sm + Nd)_N$

$\delta Ce = (Ce)_N / 0.5(La + Pr)_N$

$Ce_{anom} = \log(Ce/Ce^*)_{SN} = \log[\,2Ce_{SN}/(La_{SN} + Pr_{SN})\,]$

**Supplement Table 1 Geochemical character parameters of Carbonate rocks. C1-C4 represent the gray sediment infilled in the burrows, C5-C8 represent the white carbonate and M1 represent the thin black ferromanganese crusts, some white part may be mix unavoidable.**

---

## Author Comment (AC2) · 27 May 2018

RC1: This manuscript evaluates the impact of macrofaunal activity on the lithification of deepsea carbonates. This appears to be an interesting and previously not very thoroughly explored topic, and has the potential to advance our understanding of both the impact of bioturbation, as well as the controls on carbonate lithification. I especially appreciate the fact that this paper is very to the point, and not overly extended (although it does require some in depth discussion on some points, see below).

Reply: Thank you very much for your appreciation on the overall performance of the research work.

[Figure]

RC1: One immediate problem with the manuscript in its current form is the sometimes confusing writing style.

Reply: We took the utmost care to refine our English in the revised version.

RC1: Additionally, most of the discussion rests on speculation, rather than data, which undermines the value of this manuscript. I believe the authors should expand their observations, and especially include a thorough, quantitative assessment to elucidate the relative importance of bioturbation for carbon lithification.

Reply: Thank you for your comments on the discussion part. We have made efforts to improve the discussion basing on your suggestions. In revised manuscript, (1) CT images are further analyzed by line scanning of pixel values to play with the contrast of density change around the burrow; (2) Geochemical evidence that indicate the diagenetic differences of different types of samples are discussed thoroughly. To be fair, pore water chemical data around the burrow are valuable to quantitative assessment of bioturbation and carbonate lithification. Nevertheless, carbonate samples in this study were collected by TV-grabs bucket, which were too difficult to take the measurement of pore water chemistry. Thus, discussions of lithification enhanced by bioturbation are based on the evidence from CT images, microstructures and geochemical data.

Major specific comments: RC1: Language: While this is not the case for the manuscript as a whole, there are a lot of parts which suffer from bad spelling and grammar (see the technical corrections for specific examples, that list is not exhaustive). Before this manuscript can be accepted, it should be thoroughly revised for language.

Reply: Thank you very much for your reminding in English language and specific corrections. We have deeply checked for English language.

RC1: Interpretation of the figures: I have some difficulties with following the interpretation of the images presented in Fig. 2 and 3. On P7L6-7 you state that it is difficult to know the real depth of each burrow. How exactly do you then go from the pictures in

Fig.2a-d to the burrow shapes in Fig. 2e? How do you know they are J-shaped, and not just U-shaped, but broken off? Did you do this by eye, or did you use a cast, or scanning techniques?

Reply: Sorry for the confusion. In revised manuscript, burrow shape and Fig 2e are interpreted and discussed in detail. The depth and shape of burrow are summarized form close inspection of CT images. Especially with the help of CT analysis, we can classify burrows in straight branched or J- and U-shaped (Fig r1). If the J-shaped burrow is mistaken by broken U-shaped, it can be showed by the symmetrical difference of density distribution. The evidence that no density change symmetrically (Fig r1c, Yellow arrow), make it classified as J-shaped burrow. Based on the CT images, we give the sketch for different burrow structures.

RC1: Please expand the same remark for Fig. 3, you state (P5L22-23) that 'the most readily observable feature is the localized enhancement of density around the boring'. I have looked at the images, and even with the arrows, I have a lot of difficulties finding these enhancements. Since most of your discussion rests on these observations, I think you should expand and more clearly explain on what this statement is based. You might want to consider playing with the contrast, or other visual techniques, to make these features more clear (cause now I cannot see them).

Reply: Thanks for your advice. We have used multiple image analytical approaches to make these features more clear. We pick the pixel values of the CT image to contrast the changes of density around burrows. It is showed by the line scan profile that pixel values around the bioturbated area are higher than the matrix. This evidence supports the localized enhancement of density around burrows (Fig 3d). 3D reconstruction of the sample by CT shows more visible density contrast (Fig 4c).

RC1: Additionally, you need to expand on the statistics used to generate the linear correlations in Fig. 5.

Reply: Function of "polyfit" in MATLAB is used to estimate the difference between

bioturbated area and the control with 95% confidence intervals. (Fig. 5). Function "polyfit" returns a polynomial p(area) that is best fit for the data of integrated density. With 95% confidence bounds, the functions of bioturbated area and control are discrete with a statistical significance.

RC1: Discussion: I believe the biggest problem with this manuscript is the lack of data, and a quantitative discussion. I am supportive of the authors efforts, and I do believe that bioturbation could play a role in carbonate lithification. However, to make this case based on a few images, without any quantitative discussion, seems a bit of a short-cut. The authors should expand on these observations, and give a better mechanistic explanation on how bioturbation would enhance lithification, and include a thorough quantitative assessment of the relative importance of this process.

Reply: Thank you very much for your comments. In revised manuscript, we take the comparison of density change from CT images by line scanning of pixel values. Area near the burrow always show higher pixel value than area away far from the burrow (Fig 3d). The evidence that enhanced density near the burrow supports our deduction that carbonate lithification is enhanced by bioturbation. Enhanced lithification is also supported by micro-structure of nanofossil. Furthermore, geochemical records including elemental and stable isotopic results indicate the lithification influenced by bioturbation. Organic matter re-distribution by bioturbation acceleration the microbial oxidation around the burrow. If pore water profile around the burrow is obtained, more quantitative assessment of bioturbation and carbonate lithification can be done. Nevertheless, carbonate samples in this study were collected by TV-grabs bucket, which is too difficult to take the measurement of pore water chemistry.

Minor specific comments: RC1: The authors use 'boring' throughout the manuscript when they discuss the burrowing of macrofaunal organisms. This is rather confusing, as in my experience, it is more common to use 'burrowing' and 'burrow(s)'. I would suggest to change this throughout the manuscript to improve readability

Reply: Thanks for your advice. In revised manuscript, boring are replaced by burrow or bioturbation.

RC1: Additionally, I had to google the word 'endolith'. While it is correctly used, I again would suggest to avoid the use of this word, and use the more common 'macrofaunal', 'benthic fauna' or others, to improve readability

Reply: It has been changed. The revised title is "Macrofauna bioturbation Enhance the Deep-sea Carbonate Lithification on the Southwest Indian Ridge". "endolith" used in the manuscript are also changed.

RC1: Results: at the start of the results (P5L3) it is immediately stated that the macro-faunal burrows were lined with ferromanganese crusts. While this assumption is used aplenty throughout the manuscript (see for example Fig. 7), you do not provide evidence that these are indeed ferromanganese crusts. Please justify this assumption (or say it is just an assumption, but then explain why).

Reply: They are named basing on the elemental composition. In revised manuscript, "Mn- and Fe-oxide precipitates" are used instead of "Ferromanganese crusts" because that "Mn- and Fe-oxide precipitates" is a better term to describe our samples. Elemental composition of Mn- and Fe-oxide precipitates by SEM-EDS Element C O Na Mg Al Si Cl Ca Mn Fe Total Wt % 3.9 19.44 0.93 3.19 1.95 2.63 2.13 42.64 18.89 4.31 100 Elemental composition of chalk by SEM-EDS Element C O Mg Al Si Ca Total Wt % 3.81 19.5 1.49 1.47 2.14 71.59 100

RC1: P6L21-25: a lot of discussion about Sr, but it is not shown? Table 1 is too long to be readable. I would suggest to make this a supplementary table, and take out the most important trends and plot those in a figure.

Reply: It is visible to plot the important trend (e.g. Sr/Ca) in a figure and show elemental data in a supplementary table.

RC1: P6L33: how exactly does bioleaching deplete the isotopic value?

Reply: Stable isotope values of carbonate rock reflect a mixture of calcareous biogenic debris which is equilibrium with sea water during the growth of organisms and the alteration of diagenetic fluid. One of mechanism that bioturbation can enhance the carbonate lithification is the microbial oxidation of organic matter increasing the pore-water $CO_2$ concentration. Microbial metabolic reaction usually leads to enrichment of biospheric carbon in $^{12}C$. Thus, the dissolution and reprecipitation of carbonate influenced by bioturbation could enrich in heavy carbon in the interior of carbonate. Meanwhile, light carbon enriched in newly burrowing portion like the gray excrements. This has been further discussed in revised manuscript.

RC1: P7L15-16: why do these carbonate deposits form a favourable environment? What is special about them?

Reply: Sorry for misleading. This sentence has been deleted.

RC1: P7L5-11: I cannot follow the reasoning behind this estimation. You cannot find the real depth of the burrows, but assume 6 cm, with is the median value. How do you get a median value if you cannot determine the real burrow depth? How do you get to 12 holes per 1dm2 surface? Should you not compare volume to volume? Please be more explicit.

Reply: Sorry for the confusion. We want to estimate the volume of burrows occupied in carbonate samples. So a comparison of volume to volume is used. Although real depth of burrow is hardly to measure, we can make the estimation from the CT image. CT images are helpful to peer inside the carbonates. For one burrow in different CT slides, the slides with longest size is taken to estimate the burrow length (Fig r3 left). At the same time, the density of burrow is also enumerable from the CT images.

Technical corrections: RC1: - P1L11: 'macrofaunal inhabitants' -> not correct, better: 'benthic macrofauna'

Reply: Thanks for your correction. We have checked through the manuscript.

RC1: - P1L15-16: 'Our study reports an unfamiliar phenomenon : : : and interested by the : : :' -> this sentence is very vague, and also wrong (what is interested?), please rephrase

Reply: Thanks for your reminding. It has been rephrased. Here, we report the lithification of deep-sea carbonate associated with macrofunal burrowing.

RC1: - P1L16-17: 'These carbonate rocks may : : :' -> it is not the carbonate rocks that provide a mechanism, please rephrase: : :

Reply: It has been rephrased. Macrofaunal burrowing provides a novel driving force for deep-sea carbonate lithification at the seafloor, illuminating the geological and biological importance of deep-sea carbonate rocks on global mid-ocean ridges.

RC1: - P1L29: 'remains' -> remain

Reply: Thanks for your correction.

RC1: - P2L10: 'Burrowing and boring' -> I believe these are synonyms 'because it enhances'-> because they enhance

Reply: The sentence has been rephrased. Benthic fauna drilling into the substrate play a critical role in sediment evolution

RC1: - P2L13: 'organismic burrowing and boring' -> same remark as above, and organismic can be removed

Reply: Thanks for your correction.

RC1: - P21L21: 'between the bioturbation' -> 'between bioturbation'

Reply: Thanks for your correction.

RC1: - P2L30: 'it has been well proved' -> it has been well proven 'bursting' -> what does this sentence mean? Biogenic bloom was bursting?

Reply: The sentence has been rephrased. It has been widely reported that primary

productivity increased substantially at the Indian Ocean during the Latest Miocene–Early Pliocene

RC1: - P2L32-P3L2: I understand the sentence, but he is not constructed correctly : : :

Reply: The sentence has been rephrased. This phenomenon known as "biogenic bloom" promoted significantly high quantities of carbonates deposit at the seafloor between 9 to 3.5Ma.

RC1: - P5L9: 'herald' -> indicate

Reply: Thanks for your correction.

RC1: - P5L10-11. What does this sentence mean?

Reply: We have rephrased in a comprehensible way. Burrows can be classified in three categories.

RC1: - P5L14-16: the message you are trying to convey is unclear, please rephrase

Reply: The sentence has been rephrased. It has been suggested that Mn- and Fe-oxide precipitates grow at very slow rate of 1-10mm/Ma. Coating of black Mn- and Fe-oxide precipitates on the surface of the latter two burrows indicate that they may form much earlier than other burrows.

RC1: - P5L31 'quart' -> quartz?

Reply: Thanks for your correction.

RC1: - P6L13-15: sentence does not make much sense

Reply: The sentence has been rephrased. It is common to observe the accretionary overgrowth of calcite around the foraminifera test form SEM image (Fig. 6c). Dissolution of the coccolith plates is evident both on the surface of the thin black Mn- and Fe-oxide precipitates and in the interior of carbonate rocks (Fig 6e)

RC1: - P6L17: 'dipartite evolutionary of diagenesis' -> what does this mean?

Reply: This sentence has been rephrased. Smooth surfaces of the coccoliths in gray excrements reveal that dissolution commonly occurs influenced by bioleaching of benthic fauna (Fig 6f).

RC1: - P6L21-25: 'character' -> characteristic, 'is highly variable of Sr' -> is the highly variable Sr, 'different portion of' -> different portions of, 'mainly accounted for the substitution' -> mainly caused by substitution 'recrystallization, resulting in' -> recrystallization results in 'The loss of' -> the decrease of 'could also a response' -> could also be a response

Reply: Thanks for your correction. This paragraph has been carefully revised. Three types of samples (chalk, gray excrements and thin black Mn- and Fe-oxide precipitates) exhibit similar elemental concentration patterns for high CaO content, reflecting the strong dilution effect of biogenic calcium. One of the main characteristics of major and rare elements is the highly variable Sr concentrations in different portions of the carbonate. The storage of Sr on seafloor is mainly caused by substitution of Ca in calcium carbonate while the diagenetic recrystallization results in the decrease of Sr from the sediment (Plank and Langmuir, 1998; Qing and Veizer, 1994). The lower of Sr/Ca in chalk compared to the gray excrements could also be a response to the lithification of carbonate (Fig 7). Although biogenic calcium diluted the detrital REE fraction, it made little direct contribution to bulk REE concentrations (Xiong et al., 2012). REE patterns of the three types of sample do not exhibit any hydrothermal anomalies, e.g. positive Eu anomaly, but inherit the characteristics of sea water by enrichment of HREE compared with LREE and negative Ce anomaly (except the Mn- and Fe-oxide) (Fig. 8). The influence of nearby hydrothermal system and other detrital input to the studied carbonate area should be negligible during the lithification history.

RC1: - P6L31: statement needs a reference ('typical values for biogenic carbonates')

Reply: A classical references was added.

RC1: - P7L12 'several boring purposes are served for the benthic animals' -> does not make any sense, benthic fauna form burrows for certain purposes.

Reply: Thanks for your reminding. It has been rephrased. Benthic fauna form burrows for certain purposes of gaseous exchange, food transport, gamete transport, transport of environmental stimuli, and removal of metabolites.

RC1: - P8L1-2: 'Alternatively, bacteria and organic detritus are considered to the major source of benthic fauna in deep-sea' -> this sentence means that benthic fauna originates from bacteria and organic detritus. While this is possibly true from an evolutionary perspective, I do not think this is what you want to say here : :

Reply: Sorry for misleading. It has been corrected. Alternatively, bacterial metabolites and organic detritus are considered to the major source of food for benthic fanua in deep-sea environment which is limited by availability of organic matter.

Please also note the supplement to this comment:
https://www.biogeosciences-discuss.net/bg-2018-46/bg-2018-46-AC2-supplement.pdf

———————————————————

[Figure]

Fig. 1.

[Figure]

**Fig. 2.**

[Figure]

**Fig. 3.**

**Fig. 4.**

[Figure]

**Fig. 5.**

**Fig. 6.**

[Figure]

Fig. 7.

[Figure]

[Figure]

**Fig. 8.**

[Figure]

**Fig. 9.**

**Fig. 10.**

**Supplement:**

[revised manuscript text omitted]

$\delta Eu = (Eu)_N / 0.5(Sm + Nd)_N$

$\delta Ce = (Ce)_N / 0.5(La + Pr)_N$

$Ce_{anom} = \log(Ce/Ce^*)_{SN} = \log[\, 2Ce_{SN} / (La_{SN} + Pr_{SN})\,]$

**Supplement Table 1 Geochemical character parameters of Carbonate rocks. C1-C4 represent the gray sediment infilled in the burrows, C5-C8 represent the white carbonate and M1 represent the thin black ferromanganese crusts, some white part may be mix unavoidable.**

[Figure]

*Figure r1 CT images of chalk samples. Arrows in different colors represent different shape of burrows. Red -- Y-shaped, blue -- branched, yellow -- J-shaped and Green -- U-shaped.*

[Figure]

*Figure r2, CT slides of same carbonate sample.* Scale bar=10cm

---

## Author Response (AR1)

*Dear editor:*

*We highly appreciate the opportunity for submitting a revised version of our manuscript. We are thankful for all the valuable comments and suggestions. Here, we submit a thoroughly revised version and marked-up version of our manuscript, which has been modified according to the reviewers' suggestions.*

- *Efforts are made to check for English language and to correct typos. Co-author Dr. Dasgupta proofreads the manuscript.*
- *CT analysis and image analytical approaches, including contrast resetting (revised Fig. 3c), pixel values extraction (revised Fig. 3d), 3D reconstruction (revised Fig. 4), are used to play with the contrast of density change around the burrow. Comparison of integrated density around 113 burrows and undisturbed area are statistical analyzed with the function "polyfit" in MATLAB following reviewer's suggestion (revised Fig. 5). The functions of bioturbated zones and control zones are discrete with a statistical significance (with 95% confidence bounds).*
- *Micro-structure of nanofossils, elemental and isotopic evidences are discussed to explore the mechanism of carbonate lithification enhanced by bioturbation. Stable carbon isotope data of different portions of carbonate samples (revised table 1) are valuable to interpret the dissolution and recrystallization of carbonates. Schematic model for carbonate lithification influenced by bioturbation on the SWIR (revised Fig. 10) is highlighted and discussed to interpret the significance for their case.*

*Below we have pasted in the entire review, and we have inserted our responses to the suggestions (blue font).*

*Sincerely,*

*Xiaotong Peng, on behalf of the co-authors.*

**Response to the comments by referee#1**

**RC1**: This manuscript evaluates the impact of macrofaunal activity on the lithification of deepsea carbonates. This appears to be an interesting and previously not very thoroughly explored topic, and has the potential to advance our understanding of both the impact of bioturbation, as well as the controls on carbonate lithification. I especially appreciate the fact that this paper is very to the point, and not overly extended (although it does require some in depth discussion on some points, see below).

*Reply: Thank you very much for your appreciation on the overall performance of the research work.*

**RC1**: One immediate problem with the manuscript in its current form is the sometimes confusing writing style.

*Reply: We take the utmost care to refine our English. In revised version, co-author Dr. Dasgupta proofread the manuscript.*

**RC1**: Additionally, most of the discussion rests on speculation, rather than data, which undermines the value of this manuscript. I believe the authors should expand their observations, and especially include a thorough, quantitative assessment to elucidate the relative importance of bioturbation for carbon lithification.

*Reply: Thank you for your comments on the discussion part. We have made efforts to improve the discussion basing on your suggestions. In revised manuscript, we first demonstrate the localized changes around burrow in several aspect in section 5.1. This part is based on the close inspection of hand specimens and CT images. Then, how these changes can enhance the carbonate lithification in deep-sea is discussed in section 5.2. Geochemical evidence indicating the diagenetic differences around the burrows are discussed thoroughly. Stable carbon isotope data are used as tracers to discuss the pathways of lithification influenced by bioturbation. It is assumed that faecal pellets may strongly depleted in $^{13}C$ in isotopic mass balance with the $^{13}C$ enrichment of the organism (Damste et al., 2002). Isotopic composition in gray excrements is lighter compared to the chalks. That means bioturbated organic particles like mucus will inherit enriched $^{13}C$, which is the major carbon source for microbial metabolic reaction. Thus, the local elevated concentration of dissolved $CO_2$ in pore water trigger the dissolution of the original $CaCO_3$ phases and lead to the reprecipitation of calcites as cements with higher $^{13}C$ in carbonate rocks.*

**Major specific comments:**

**RC1**: Language: While this is not the case for the manuscript as a whole, there are a lot of parts which suffer from bad spelling and grammar (see the technical corrections for specific examples, that list is not exhaustive). Before this manuscript can be accepted, it should be thoroughly revised for language.

*Reply: Thank you very much for your reminding in English language and specific corrections. We have deeply checked for English language.*

**RC1**: Interpretation of the figures: I have some difficulties with following the interpretation of the images presented in Fig. 2 and 3. On P7L6-7 you state that it is difficult to know the real depth of each burrow. How exactly do you then go from the pictures in Fig.2a-d to the burrow shapes in Fig. 2e? How do you know they are J-shaped, and not just U-shaped, but broken off? Did you do this by eye, or did you use a cast, or scanning techniques?

*Reply: We have included detailed interpretations and comparisons of burrow shape in the revised paper version.*
*The depth and shape of burrow are summarized form close inspection of CT images. With the help of CT analysis, we can classify burrows in straight branched or J- and U-shaped (Fig S1). If the J-shaped burrow is mistaken by broken U-shaped, it can be showed by the symmetrical difference of density distribution. The evidence that no density change symmetrically (Fig S1c and d, Yellow arrow), make it classified as J-shaped burrow. Based on the CT images, we give the sketch for different burrow structures in Fig. 2e.*
*CT images are helpful to peer inside the carbonates. For one burrow in different CT slides, the slides with longest burrow size is taken to estimate the burrow length.*

[Figure]

*Figure S1 CT images of carbonate samples. b, c and d are different CT scanning slides of same carbonate rock. The slice thickness between c and d is 2.5 mm*
*Arrows in different colors represent different shape of burrows. Red -- Y-shaped, blue -- branched, yellow -- J-shaped and Green -- U-shaped.*

**RC1**: Please expand the same remark for Fig. 3, you state (P5L22-23) that 'the most readily observable feature is the localized enhancement of density around the boring'. I have looked at the images, and even with the arrows, I have a lot of difficulties finding these enhancements. Since most of your discussion rests on these observations, I think you should expand and more clearly explain on what this statement is based. You might want to consider playing with the contrast, or other visual techniques, to make these features more clear (cause now I cannot see them).

*Reply: Thanks for your advice. We agree that other visual techniques are required to make*

*the contrast more clear. In the revised manuscript version, we have used multiple image analytical approaches to make these features more clear. In Fig.3, we pick the pixel values of the CT image to contrast the changes of density around burrows. It is showed by the line scan profile that pixel values around the bioturbated area are higher than the matrix (Fig 3d). This evidence supports the localized enhancement of density around burrows. In Fig. 4, 3D reconstruction of the sample by CT analysis shows more visible density contrast (Fig 4c).*

**RC1**: Additionally, you need to expand on the statistics used to generate the linear correlations in Fig. 5.

*Reply: Data used for comparison come from the gray values of CT images. In the revised manuscript, function of "polyfit" in MATLAB, which is suggested by anonymous Referee #2, is used to estimate the difference between bioturbated area and the control with 95% confidence intervals. (Fig. 5). Function "polyfit" returns a polynomial p(area) that is best fit for the data of integrated density. With 95% confidence bounds, the functions of bioturbated area and control are discrete with a statistical significance.*

**RC1**: Discussion: I believe the biggest problem with this manuscript is the lack of data, and a quantitative discussion. I am supportive of the authors efforts, and I do believe that bioturbation could play a role in carbonate lithification. However, to make this case based on a few images, without any quantitative discussion, seems a bit of a short- cut. The authors should expand on these observations, and give a better mechanistic explanation on how bioturbation would enhance lithification, and include a thorough quantitative assessment of the relative importance of this process.

*Reply: Thank you very much for your comments. In revised manuscript, we take the comparison of density change from CT images by line scanning of pixel values. Area near the burrow always show higher pixel value than area away far from the burrow (Fig 3d). The evidence that enhanced density near the burrow supports our deduction that carbonate lithification is enhanced by bioturbation. What's more, enhanced lithification is also supported by micro-structure of nanofossil. In addition, geochemical records including elemental and stable isotopic results indicate the lithification influenced by bioturbation. Organic matter re-distribution by bioturbation acceleration the microbial oxidation around the burrow.*

**Minor specific comments:**

**RC1**: The authors use 'boring' throughout the manuscript when they discuss the burrowing of macrofaunal organisms. This is rather confusing, as in my experience, it is more common to use 'burrowing' and 'burrow(s)'. I would suggest to change this throughout the manuscript to improve readability

*Reply: We highly appreciate the term names clarification, and understand that is preferable avoid any confusing terminology. In revised manuscript, "boring" are replaced by "burrow".*

**RC1**: Additionally, I had to google the word 'endolith'. While it is correctly used, I again would suggest to avoid the use of this word, and use the more common 'macrofaunal', 'benthic fauna' or others, to improve readability

*Reply: It has been changed. The revised title is "Macrofauna bioturbation Enhance the Deep-sea Carbonate Lithification on the Southwest Indian Ridge". "Endolith" used in the manuscript are also changed.*

**RC1**: Results: at the start of the results (P5L3) it is immediately stated that the macrofaunal burrows were lined with ferromanganese crusts. While this assumption is used aplenty throughout the manuscript (see for example Fig. 7), you do not provide evidence that these are indeed ferromanganese crusts. Please justify this assumption (or say it is just an assumption, but then explain why).

*Reply: They are named basing on the elemental composition. In revised manuscript, "Mn- and Fe-oxide precipitates" are used instead of "Ferromanganese crusts" because that "Mn- and Fe-oxide precipitates" is a better term to describe our samples.*

Elemental composition of Mn- and Fe-oxide precipitates by SEM-EDS

| Element | C | O | Na | Mg | Al | Si | Cl | Ca | Mn | Fe | Total |
|---|---|---|---|---|---|---|---|---|---|---|---|
| Wt % | 3.9 | 19.44 | 0.93 | 3.19 | 1.95 | 2.63 | 2.13 | 42.64 | 18.89 | 4.31 | 100 |

Elemental composition of chalk by SEM-EDS

| Element | C | O | Mg | Al | Si | Ca | Total |
|---|---|---|---|---|---|---|---|
| Wt % | 3.81 | 19.5 | 1.49 | 1.47 | 2.14 | 71.59 | 100 |

**RC1**: P6L21-25: a lot of discussion about Sr, but it is not shown? Table 1 is too long to be readable. I would suggest to make this a supplementary table, and take out the most important trends and plot those in a figure.

*Reply: The change was done accordingly. It is visible to plot the important trend (e.g. Sr/Ca) in a figure and show elemental data in a supplementary table.*

**RC1**: P6L33: how exactly does bioleaching deplete the isotopic value?

*Reply: Stable isotope values of carbonate rock reflect a mixture of calcareous biogenic debris which is equilibrium with sea water during the growth of organisms and the alteration of diagenetic fluid. One of mechanism that bioturbation can enhance the carbonate lithification is the microbial oxidation of organic matter increasing the pore-water $CO_2$ concentration. Microbial metabolic reaction usually leads to enrichment of biospheric carbon in $^{12}C$. Thus, the dissolution and reprecipitation of carbonate influenced by bioturbation could enrich in heavy carbon in the interior of carbonate. Meanwhile, light carbon enriched in newly burrowing portion like the gray excrements. This has been further discussed in revised manuscript.*

**RC1**: P7L15-16: why do these carbonate deposits form a favourable environment? What is special about them?

*Reply: Sorry for misleading. This sentence has been deleted.*

**RC1**: P7L5-11: I cannot follow the reasoning behind this estimation. You cannot find the real depth of the burrows, but assume 6 cm, with is the median value. How do you get a median value if you cannot determine the real burrow depth? How do you get to 12 holes per 1dm2 surface? Should you not compare volume to volume? Please be more explicit.

*Reply: Sorry for the confusion. We want to estimate the volume of burrows occupied in carbonate samples. So a comparison of volume to volume is used. Although real depth of burrow is hardly to measure, we can make the estimation from the CT image. CT images are helpful to peer inside the carbonates. For one burrow in different CT slides, the slides with longest size is taken to estimate the burrow depth. At the same time, the density of burrow is also enumerable from the CT images.*

**Technical corrections:**

**RC1**: - P1L11: 'macrofaunal inhabitants' -> not correct, better: 'benthic macrofauna'

*Reply: Thanks for your correction. We have checked through the manuscript.*

**RC1**: - P1L15-16: 'Our study reports an unfamiliar phenomenon : : : and interested by the : : :' -> this sentence is very vague, and also wrong (what is interested?), please rephrase

*Reply: Thanks for your reminding. It has been rephrased.*
*"Here, we report the lithification of deep-sea carbonate associated with macrofaunal burrowing."*

**RC1**: - P1L16-17: 'These carbonate rocks may : : :' -> it is not the carbonate rocks that provide a mechanism, please rephrase: : :

*Reply: It has been rephrased.*
*"Macrofaunal burrowing provides a novel driving force for deep-sea carbonate lithification at the seafloor, illuminating the geological and biological importance of deep-sea carbonate rocks on global mid-ocean ridges."*

**RC1**: - P1L29: 'remains' -> remain

*Reply: Thanks for your correction.*

**RC1**: - P2L10: 'Burrowing and boring' -> I believe these are synonyms 'because it enhances'-> because they enhance

*Reply: The sentence has been rephrased.*
*"Benthic fauna drilling into the substrate play a critical role in sediment evolution."*

**RC1**: - P2L13: 'organismic burrowing and boring' -> same remark as above, and organismic can be removed

*Reply: Thanks for your correction.*

**RC1**: - P21L21: 'between the bioturbation' -> 'between bioturbation'

*Reply: Thanks for your correction.*

**RC1**: - P2L30: 'it has been well proved' -> it has been well proven 'bursting' -> what does this sentence mean? Biogenic bloom was bursting?

*Reply: The sentence has been rephrased.*
*"It has been widely reported that primary productivity increased substantially at the Indian Ocean during the Latest Miocene–Early Pliocene."*

**RC1**: - P2L32-P3L2: I understand the sentence, but he is not constructed correctly : : :

*Reply: The sentence has been rephrased.*
*"This phenomenon known as "biogenic bloom" promoted significantly high quantities of carbonates deposit at the seafloor between 9 to 3.5Ma."*

**RC1**: - P5L9: 'herald' -> indicate

*Reply: Thanks for your correction.*

**RC1**: - P5L10-11. What does this sentence mean?

*Reply: We have rephrased in a comprehensible way.*
*"Burrows can be classified in three categories."*

**RC1**: - P5L14-16: the message you are trying to convey is unclear, please rephrase

*Reply: The sentence has been rephrased.*
*"It has been suggested that Mn- and Fe-oxide precipitates grow at very slow rate of 1-10mm/Ma. Coating of black Mn- and Fe-oxide precipitates on the surface of the latter two burrows indicate that they may form much earlier than other burrows."*

**RC1**: - P5L31 'quart' -> quartz?

*Reply: Thanks for your correction.*

**RC1**: - P6L13-15: sentence does not make much sense

*Reply: The sentence has been rephrased.*
*"It is common to observe the accretionary overgrowth of calcite around the foraminifera test form SEM image (Fig. 6c). Dissolution of the coccolith plates is evident both on the surface of the thin black Mn- and Fe-oxide precipitates and in the interior of carbonate rocks (Fig 6e)."*

**RC1**: - P6L17: 'dipartite evolutionary of diagenesis' -> what does this mean?

*Reply: This sentence has been rephrased.*
*"Smooth surfaces of the coccoliths in gray excrements reveal that dissolution commonly occurs influenced by bioleaching of benthic fauna (Fig 6f)."*

**RC1**: - P6L21-25: 'character' -> characteristic, 'is highly variable of Sr' -> is the highly variable Sr, 'different portion of' -> different portions of, 'mainly accounted for the substitution' -> mainly caused by substitution 'recrystallization, resulting in' -> recrystallization results in 'The loss of' -> the decrease of 'could also a response' -> could also be a response

*Reply: Thanks for your correction. This paragraph has been carefully revised.*
*"Three types of samples (chalk, gray excrements and thin black Mn- and Fe-oxide precipitates) exhibit similar elemental concentration patterns for high CaO content, reflecting the strong dilution effect of biogenic calcium. One of the main characteristics of major and rare elements is the highly variable Sr concentrations in different portions of the carbonate. The storage of Sr on seafloor is mainly caused by substitution of Ca in calcium carbonate while the diagenetic recrystallization results in the decrease of Sr from the sediment (Plank and Langmuir, 1998; Qing and Veizer, 1994). The lower of Sr/Ca in chalk compared to the gray excrements could also be a response to the lithification of carbonate (Fig 7). Although biogenic calcium diluted the detrital REE fraction, it made little direct contribution to bulk REE concentrations (Xiong et al., 2012). REE patterns of the three types of sample do not exhibit any hydrothermal anomalies, e.g. positive Eu anomaly, but inherit the characteristics of sea water by enrichment of HREE compared with LREE and negative Ce anomaly (except the Mn- and Fe-oxide) (Fig. 8). The influence of nearby hydrothermal system and other detrital input to the studied carbonate area should be negligible during the lithification history."*

**RC1**: - P6L31: statement needs a reference ('typical values for biogenic carbonates')

*Reply: A classical references was added.*

**RC1**: - P7L12 'several boring purposes are served for the benthic animals' -> does not make any sense, benthic fauna form burrows for certain purposes.

*Reply: Thanks for your reminding. It has been rephrased.*
*"Benthic fauna form burrows for certain purposes of gaseous exchange, food transport, gamete transport, transport of environmental stimuli, and removal of metabolites."*

**RC1**: - P8L1-2: 'Alternatively, bacteria and organic detritus are considered to the major source of benthic fauna in deep-sea' -> this sentence means that benthic fauna originates from bacteria and organic detritus. While this is possibly true from an evolutionary perspective, I do not think this is what you want to say here : :

*Reply: Sorry for misleading. It has been corrected.*
*"Alternatively, bacterial metabolites and organic detritus are considered to the major source of food for benthic fanua in deep-sea environment which is limited by availability of organic*

*matter."*

**. Response to the comments by referee#2**

**RC2**: The manuscript titled "Endolithic Boring Enhance the Deep-sea Carbonate Lithification on the Southwest Indian Ridge" details observations and analyses of deep-sea carbonate samples that appear to be experiencing enhanced lithification associated with benthic faunal burrowing. The study employs computed X-ray tomography, visual and microscope observation, and geochemistry to evaluate the relationships between burrowing and the degree of carbonate lithification. The main conclusion is that burrowing is likely an important process accelerating carbonate lithification in the deep-sea. The findings are intriguing and certainly of interest to a wide readership.

*Reply: We are very thankful to the anonymous reviewer for constructive feedbacks and insightful comments on our manuscript.*

**RC2**: My main reservations about this manuscript are twofold: (1) it is not immediately clear in some CT-scan images that there are density contrasts (enhanced lithification) surrounding the burrows (Figure 4).

*Reply: We agree that the CT images in earlier version of this manuscript need improvement. In order to make the density contrasts clearly, we pick the pixel values of the CT image to contrast the change of density around burrow. It is showed by the line scan profiles that pixel values around the bioturbated area is higher than matrix indicating the localized enhancement of density around burrows (Fig 3d). 3D reconstruction of the sample by CT analysis also has been added in revised manuscript (Fig 4b). It can conclude that the enhancement of density around the burrows is consistent with the halo defined by the sediment being lighter in color (Fig 4a, c).*

**RC2**: (2) it is not clear based on the data treatment that there is true statistical significance in the difference in density between bioturbated zones and control zones (Figure 5). See specific comments on these below. If the authors can address the above major points then I can see this manuscript being of interest to a wide readership. I agree with the authors that burrowing-enhanced lithification would appear to be an important process if it can extrapolated to deep-sea carbonates world-wide.

*Reply: Thanks very much for your advisable suggestions to promote the quality of data treatment. In revised paper, function "polyfit" in MATLAB as you advised is used to generate the polynomial p(area) that is a best fit for the data for integrated density (with 95% confidence bounds). The functions of bioturbated zones and control zones are discrete with a statistical significance (Fig 5).*
*Data used for comparison come from the gray values of CT images. In Fig. 3d, an example of the density change around the burrow is showed by line scanning. When comes to the Fig 5, whose data are generated from 113 burrows, the statistical results support our conclusion that macrofaunal burrowing enhance the deep-sea carbonate lithification on the Southwest Indian*

*Ridge.*

**RC2**: While the English is already commendable for authors for whom English might be a second language, and it is possible to follow what the authors are saying throughout the manuscript, there remain minor issues with English throughout the manuscript. This should be easily fixed with a careful proofread by a native speaker.I would consider the revisions required to address the general comments above and specific comments below to be major - significant blocks of text should be revised and additional statistical treatment should be applied to the dataset.

*Reply: Thank you very much for your reminding in English language. In revised version, co-author Dr. Dasgupta proofread the manuscript.*

Specific Comments
**RC2**: Abstract, line 9: I'm not sure that one can say that lithification of deep-sea carbonates is a "mystery"; there is a respectable body of literature on lithification mechanisms and rates dating back over three decades. Perhaps better would be something like "the role of deep-sea macrofauna in their lithification remain poorly understood".

*Reply: The sentence has been revised.*
*"The role of deep-sea macrofauna in carbonate lithification remains poorly understood."*

**RC2**: Abstract, line 12: "in the sample" makes it read as if there was only a single hand sample, when it appears that grab buckets provided multiple samples. This occurs elsewhere in the manuscript as well.

*Reply: Thanks for reminding. It has been modified and we have checked the whole manuscript to avoid this mistake.*

**RC2**: Abstract line 16: "interested by" doesn't make much sense - please re-phrase.

*Reply: The sentence has been deleted.*

**RC2**: Abstract, last sentence: these results don't really speak to the importance of deep-sea carbonate sediments, simply the mechanisms of their formation. Please re-phrase.

*Reply: The sentence has been revised.*
*"Macrofaunal burrowing provides a novel driving force for deep-sea carbonate lithification at the seafloor, illuminating the geological and biological importance of deep-sea carbonate rocks on global mid-ocean ridges."*

**RC2**: Main text in general: while I find that the text is written in a clear and straightforwards manner, there remain minor grammatical errors throughout. If english is the authors' second language, then they should be commended - this manuscript already reads decently well. Nonetheless, further editing by a native english speaker is necessary to wrap up the grammatical loose ends that are apparent throughout the manuscript.

*Reply: whole manuscript has been deeply checked for English language.*

**RC2**: Page 2, line 24: it might offend researchers in diverse fields to say that the entire Indian Ocean is "poorly understood".

*Reply: Thanks for the comment. This sentence has been deleted.*

**RC2**: Page 2, line 30-34: grammatical issues, please re-phrase.

*Reply: It has been modified.*
*"This phenomenon known as "biogenic bloom" promoted significantly high quantities of carbonates deposit at the seafloor between 9 to 3.5Ma (Gupta et al., 2004; Dickens and Owen, 1999)."*

**RC2**: Materials and Methods: certain phrases in the methods have been reproduced wordfor-word from previous work. For example, page 4, lines 12 through 14 - these identical lines are also found in Li et al. (2014). Even if the same methodology was used for both studies, it would be prudent to re-word the text in the methods.

*Reply: The text has been re-phrased.*
*"Small fragments of the dried samples were fixed onto aluminum stubs with two-way adherent tabs, and allowed to dry overnight. They were sputter coated with gold for 2-3 minutes before being examined on a Philips XL-30 scanning electron microscope equipped with an accelerating voltage of 15kV at the State Key Laboratory of Marine Geology, Tongji University." The elemental composition of selected spots was determined by energy dispersive X-ray (EDX) analysis on the SEM with an accelerating voltage of 20 kV.*

**RC2**: Page 4, line 21: there should be no "elution" step in this technique. Also line 22, how was precision evaluated? Repeat measurements of standards? Finally, how were these measurements standardized - using multi-element solutions or by measurement of geostandards? The methods are not sufficiently detailed here.

*Reply: Sorry for my mistake. There is no "elution" step.*
*Analytical precision was monitored using the Chinese national carbonate standard, GBW04405. Conversion of measurements to the Vienna Peedee Belemnite (PDB) scale was performed using NBS-19 and NBS-18.*

**RC2**: From page 5 onwards: these are not ferromangense crusts in the strict sense of the word. Perhaps "Mn- and Fe-oxide precipitates" is a better term.

*Reply: thanks for your advice. It has been changed.*

**RC2**: Page 5, line 10: I suggest re-phrasing this sentence.

*Reply: This sentence has been re-phrased.*

*"Burrows can be classified in three categories."*

**RC2**: Page 6, line 16-17: I suggest re-phrasing.

*Reply: This sentence has been re-phrased.*
*"Smooth surfaces of the coccoliths in gray excrements reveal that dissolution commonly occurs influenced by bioleaching of benthic fauna (Fig 6f)."*

**RC2**: Page 6, line 24–25: you can't lose a ratio (but you can lower it).

*Reply: Thanks for reminding. It has been corrected.*

**RC2**: Page 7 line 1: I suggest re-phrasing.

*Reply: This sentence has been re-phrased.*
*"Positive correlation of $\delta^{13}C_{PDB}$ and $\delta^{18}O_{PDB}$ values of chalk and gray excrements (r = 0.91) reveals minor environmental influence on early lithification (Fig. 8) and bioturbation should be a critical factor during the lithification."*

RC2: Discussion in general: it would be nice if the authors could elaborate on why a decrease in carbonate saturation state (leading to dissolution) promotes lithification (as opposed to an increase in carbonate saturation state leading to precipitation).

*Reply: Thanks for your comments. Cementation after dissolution of biogenic debris is of the important process of carbonate lithification. We have made efforts to explain the dissolution and reprecipitation of calcite to cement in revised manuscript. The dissolution of carbonate in the ocean is primarily controlled by the degree of pore water undersaturation with respect to the biogenic carbonate phase. Bioturbation can redistribute the organic matter around the burrow. Thus, oxidation of organic matter will accelerate the concentration of pore water $CO_2$ leading to the undersaturation of calcites.*
*Furthermore, thin Mn- and Fe oxide precipitates prevent the rapid ion exchange between bottom water and pore water within carbonate rocks because larger grain surfaces and porosity of fine-grained poorly sorted carbonate oozes compared to Mn- and Fe oxide precipitates. The products of $CaCO_3$ dissolution may trend to diffuse toward to the interior of carbonate rocks, and lead to an enhanced $CO_3^{2-}$ ion gradient in pore water profile and ultimately promoting the reprecipitation of calcites as cements around the burrows in carbonate rocks.*

**RC2**: Also, while aerobic respiration decreases the local carbonate saturation state, sulfate reduction will increase it. Can the authors include a statement about oxygen penetration and the depth of sulfate reduction (even if it is simply based on the findings of others in similar settings)?

*Reply: Thanks for your valuable comment.*
*We cannot exclude the potential that sulfate reduction have happened in studied samples. This can be illustrated for the present study by examining the observed variations in ion*

*content of the pore water.*

*However, carbonate samples here were collected by TV-grabs bucket. It is too difficult to take the measurement of pore water chemistry. Several literatures support our discussion that pore-water $CO_2$ by oxidation of organic matter is responsible for the carbonate dissolution. (Broecker and Peng 1982; Jahnke et al. 1994; Noé et al. 2006; Croizé et al. 2013). Metabolic activity may disintegration of organic material causing dissolution of carbonate and increasing the degree of supersaturation. In the condition that bioturbation processes succeed in redistribution of organic matter around the burrow, concentration of $CO_2$ in pore water could increase. Although we could not elaborate the influence of sulfate reduction, aerobic respiration is reasonable to the decrease of carbonate saturation state.*

**RC2**: Figure 1 Legend: The legend indicates that the red triangle is an inactive hydrothermal field while the caption indicates that it is active - this contradiction needs to be resolved. Also at the end it should read "red circle".

*Reply: It has been corrected. The red circle is active hydrothermal field and the red triangle indicates inactive fields.*

**RC2**: Figure 2e should have a scale bar.

*Reply: The scale bar has been added.*

**RC2**: Figure 4b: Contrary to the caption, it is difficult to see any enhanced of density in this image. Figure 4c and d: what do the different arrows represent? In a related vein, for Figures 4 b, c, and d in general - the areas of higher density are not obvious at all. Perhaps circle them or find some better way of highlighting these areas? Also could another presentation method be employed (e.g., an additional panel with contrast adjustments to better show the differences, perhaps shown alongside an un-modified version of the same figure for traceability)?

*Reply:   Thanks for your advice.*
*Both Fig. 3 and Fig. 4 have been changed. We pick the pixel values of the CT image to contrast the changes of density around burrows. It is showed by the line scan profiles that pixel values around the bioturbated area is higher than matrix indicating the localized enhancement of density around burrows.*

Figure 5: This is not a statistical analysis in the sense that it does not provide any measure of confidence in the comparison between the two slopes (e.g. whether they can be considered different with 95% confidence). For this you would need to use something like the function "polyfit" in MATLAB (for example). No statistical evidence is presented that these slopes are indeed different... this is a major point as the paper hinges on the importance of burrowing effects.

*Reply: Thanks very much for your valuable suggestions. The figure has been revised. Difference can be showed with the function "polyfit" with 95% confidence intervals.*

**RC2**: Figure 8: As a Kiel carbonate device was used, these are not "bulk" C isotope measurements, but C_carb measurements (same for oxygen isotopes). That is to say, organic matter in the sample is not measured during the analyses when a Kiel carbonate device is used, only carbonate - this should be clarified.

*Reply: Sorry for the mistake. It has been corrected.*

[revised manuscript text omitted]
 dimensions of boringburrow holes can be estimatedassessed by from CT analysis tomographic cross-section of the samples. to estimate the extent of substratum reconstruction. BoringBurrow holes were generally with severalfew millimetersmillimetres to 2 cm in diameter, commonly penetrating 6 to 10 cm into the chalkrocks and ultimately reaching a density of up to 12 per dm². If boring holeburrows in

5   straight, branched, or J- and U-shaped (Fig. 2e) are simplified to a cylinder with the diameter of 1 cm and height of 6 cm, which are the median value of the boring holeburrows, estimated the volume occupied by the boring can be estimated in athis simplifiede modelof the simplified cylinder would be helpful to reckon the extent of substratum reconstruction by bioturbation. The boring in straight, branched, or J- and U-shaped (Fig 2e) are simplified to a cylinder with the diameter of 1 cm and height of 6 cm, which are the median value of the boring holes. In this model, 1 dm² surface area which can harbor

10  12 boring holeburrows on the surface may reach to 0.226 dm³ boringburrow space. Eventually, Although it is hard to determine the dimension of each boring hole accurately, Aan important conclusion can be we can deduced from thise simplified model that the carbonate substratum is were reconstructed by the boring bioturbation to a great extent.

Several bBoring purposes enthic fauna maintainform boringburrows for certain purposes of are served for the benthic animals such as gaseous exchange, food transport, gamete transport, transport of environmental stimuli, and removal of

15  metabolites (Kristensen and Kostka, 2013). Polychaetes, the most successful burrow class for example (Díaz-Castañeda and Reish, 2009), were abundant and conventionally produced J- or U- shaped burrow extended as long as several decimetres in hand specimens (Fig. 2 c, e). Relic burrows allow sea water to directly penetrate into carbonate rocks, which is beneficial to the precipitation of black Mn- and Fe-oxide precipitates on the inner surface of burrow (Fig. 2a, c). The genetic models for Mn- and Fe-oxide precipitates has been attributed to the minerals precipitated out of the cold ambient seawater onto the rock

20  surface with the aid of biogenic activity (Hein and Koschinsky, 2014). Burrowing benthic fauna excrete mucus to garden their burrow holes by incorporating organic matter into the walls (Dworschak et al., 2006; Koller et al., 2006). The mucus layer may act as favourable site for the accumulation of metallic ions through organo-metallic complexation or chelation at suitable Eh, pH and redox conditions (Lalonde et al., 2010; Banerjee, 2000). Thus, along with Carbonate mass accumulation during the Latest Miocene Early Pliocene at Indo Pacific provided the bioclastic deposition to the sea floor (Singh et al.,

25  2012; Rai and Singh, 2001; Gupta et al., 2004; Arumugm et al., 2014). These carbonate sediments deposit on the seafloor thus form a favorable environment for benthic fauna (fig 2). Polychaetes, taking the most successful boring class for example (Díaz Castañeda and Reish, 2009), are abundant and conventionally produce J- or U- shaped boring extended as long as several decimeters in carbonate sample (Fig. 2 c, e). During the frequent construction and maintenance of boring, not only the carbonate reworking and bio-mixing occurs by boring during frequent construction and maintenance of burrow, the

30  redoxmineralogical and geochemical parameters, including pH, were are also assumed to oscillate around the bioturbate structureburrow-. 
[revised manuscript text omitted]

---

## Referee Report (RR1)

**Second review of BG-2018-46**

I have now completed a second review of the manuscript now re-titled "Macrofaunal burrowing Enhance Deep-sea Carbonate Lithification on the Southwest Indian Ridge".

The authors have taken the previous round of reviewer comments seriously and the manuscript is much improved. That said, there are still two outstanding issues that were raised in the original review that prevent acceptance in its current form.

First, many minor grammatical errors remain – I have made suggestions for correcting some of the most glaring errors below in the specific comments. I appreciate that special attention has been made to the English in this current revision, and it is indeed improved. However, it still requires further proof-reading for English before it can be considered ready for print. I implore the authors to consider an external proofreading service or other solution that will ensure that the grammer is up to par.

The second and more important issue relates to the statistical analyses of the change in density surrounding the burrows. While the authors provide R-squared values for their fits, they do not plot the fits, nor their confidence bounds. Instead the current figure simply has lines connecting the points, however the points are not in increasing order, so the lines bounce all around in a zig-zag manner. What I am hoping to see is the regression line drawn through the data along with the confidence bounds. See https://www.mathworks.com/help/stats/polyconf.html for an example where both the fit line and confidence bounds are shown overlaid upon the data. The basic statistical question is not "does each slope have a robust fit", which is what the authors currently provide. Instead, the basic statistical question at hand is "are the two slopes significantly different"? This is at the heart of the manuscript – whether bioturbation has a statistically significant influence on density (and thus carbonate lithification).

I don't think these should be prohibitively difficult for the authors to address, and I continue to hold the opinion that this work should be eventually suitable for publication in *Biogeosciences*. In my opinion we aren't there yet – but definitely getting closer.

Specific comments

Pg. 1, Line 10: "blanketing the seafloor of the"

Pg. 1, Line 12: "in this carbonate lithified area"

Pg. 1, Line 13: "were examined" … also "enhances".

Pg. 2, Line 18: This is a bit awkward, I recommend "We examined this intriguing occurrence of non-burial carbonate… and highlight the interactions…"

Pg. 2, Line 28: "substantially in"

Pg. 3, Line 19: "which is a public"

Pg. 3, Line 27: "The MATAB function polyfit was used"

Title of section 5.2: "around burrows"

Pg. 5, lines 29–31: There are multiple grammatical problems in this sentence.

Pg. 6, Line 4: "from SEM images"

Pg. 6, Line 20: "hydrothermal systems and detrital input"

Pg. 7, Line 24: "bulk samples are"

Pg. 7, Line 31: "deep-sea environments"

Pg. 8, Line 7: "of the studied carbonate area"

Pg. 8, Line 9: "phases", then line 10: "this is not likely to occur here"

Pg. 8, Line 12: "However, the carbonate samples studied here have never been buried"

Pg. 8, Line 15: "Moreover, ecological niches"

Pg. 8, Line 18: The beginning of this paragraph has grammatical problems.

Conclusion: Multiple grammatical issues.

Figure 5: There are lines connecting the points that show a zig-zag pattern as they trace the order of the points without any sorting. In other words, they are simply connecting the points in a random order. These lines should be removed and the actual fits presented (the trendlines that go through these point clouds, which are not currently shown).

---

## Author Response (AR2)

**Editor:** Thank you for submitting a revised manuscript. Two reviewers have now provided comments on this revised version. While both reviewers find the manuscript greatly improved, there are still a number of smaller and larger issues to address prior to publication.

The first larger issue is the English grammar, which should be improved further. While all BG-manuscripts are edited for English in the final stage before publication, a higher initial level is required. Both reviewers provide useful suggestions. If possible, please involve a native English speaker in checking the manuscript. Please note that the title also contains an error: "Enhance" should be replaced by "Enhances". The second larger issue refers to the statistics and presentation of the regression lines in Fig. 5, as pointed out by one the reviewers.

*We appreciate the work of the editor and two anonymous reviewers. We are thankful for all the valuable comments and suggestions. English languages are edited with the help from a commercial language service. Fig.5 is revised following the valuable suggestions from the reviewers. Below we have pasted in the entire review, and we have inserted our responses to the suggestions (blue font).*

**RC1:** The authors have greatly improved the manuscript and representation of the figures, I just have a few more minor comments, and some technical corrections. The only thing that bothers me a bit is the amount of grammar mistakes and spelling errors in the manuscript. I would strongly advice to proofread the MS to remove as much of these as possible (I have highlighted a few in the technical comments section.

*Thank you very much for your appreciation on the overall performance of the work.*

*Sorry for our poor English writing. We have called for Language service for help with English language editing.*

Minor specific comments:

P1L16: you are more highlighting the importance of bioturbation, not really the importance of deep-sea carbonate rocks

*The sentence has been rephrased:*

*"Macrofaunal burrowing provides a novel driving force for deep-sea carbonate lithification at the seafloor, illuminating the geological and biological importance of bioturbation on global deep-sea carbonate rocks."*

P3L5: did you take any pictures or determined what animals were exactly present in the rocks? I see now you did this in Figure 2, please add it in the method description

*It was unfortunately that we did not take pictures of burrowing animals on the sea specifically. When carbonate samples were spread on the deck, benthic organisms were usually evident among the fractured rocks. Fig. 2c, d were taken from the fractured rocks on the deck.*

P8L14: How exactly does your element and isotope results reveal minor external impact on the lithification?

*REE patterns of the three types of sample did not exhibit any hydrothermal anomalies, e.g. positive Eu anomaly, but inherit the characteristics of sea water by enrichment of HREE compared with LREE and negative Ce anomaly (except the Mn- and Fe-oxide) (Fig. 8). Positive correlation of $\delta^{13}C_{PDB}$ and $\delta^{18}O_{PDB}$ values of chalk and gray excrements (r = 0.91) reveals minor environmental influence on early lithification (Fig. 9). Thus, we deduced that the influence of nearby hydrothermal systems and other detrital input to the studied carbonate area should be negligible during the lithification history.*

*"We deduced from elements and isotope results that the influence of nearby hydrothermal systems and other detrital input to the studied carbonate area should be negligible during the lithification history"*

P9L13: same remark as above, you did not illuminate the geological and biological importance of carbonate rocks. You illuminated the importance of burrowing animals on the lithification of carbonate rocks

*The sentence has been rephrased:*

*"The novel mechanism proposed here for non-burial carbonate lithification at the deep-sea seafloor sheds light on the potential interactions between deep-sea biota and sedimentary rocks, and also illuminate the geological and biological importance of bioturbation on global deep-sea carbonate rocks."*

Technical corrections:

P1L10: a tremendous amount of burrows

*Thanks for your correction.*

P1L10: the ultraslow spreading Southwest

*Thanks for your correction.*

P1L13: enhances deep-sea carbonate lithification

*Thanks for your correction.*

P1L24: may include diagenetic products

*Thanks for your correction.*

P2L8: Benthic fauna burrowing into the substrate plays a critical role

*Thanks for your correction.*

P2L10: you could also cite more recent papers here, e.g., (doi:10.1007/s10498-016-9301-7)

*Thanks for your reminding.*

P2L29-30: promoted deposition of high quantities of carbonate deposits

*Thanks for your correction.*

P3L7: were kept frozen in dry ice

*Thanks for your correction.*

P5L29-30: this sentence is not very correct, pleas rephrase

*It has been rephrased*

*"Therefore, the carbonate deposits on the SWIR could represent bioclastic deposition from "biogenic bloom", which was the productivity related event in a large part of Indian Ocean during the middle Miocene to the early Pliocene (Singh et al., 2012; Rai and Singh, 2001; Gupta et al., 2004; Arumugm et al., 2014").*

P8L9 stable CaCO3 phases

*Thanks for your correction.*

P8L10 that is not likely to happen

*Thanks for your correction.*

P8L21: Berner and Westrich, 1985, American Journal of Science is more appropriate here

*Thanks for your reminding.*

**RC2:**

Second review of BG-2018-46

I have now completed a second review of the manuscript now re-titled "Macrofaunal burrowing Enhance Deep-sea Carbonate Lithification on the Southwest Indian Ridge".

The authors have taken the previous round of reviewer comments seriously and the manuscript is much improved. That said, there are still two outstanding issues that were raised in the original review that prevent acceptance in its current form.

First, many minor grammatical errors remain – I have made suggestions for correcting some of the most glaring errors below in the specific comments. I appreciate that special attention has been made to the English in this current revision, and it is indeed improved. However, it still requires further proof-reading for English before it can be considered ready for print. I implore the authors to consider an external proofreading service or other solution that will ensure that the grammer is up to par.

*Sorry for our poor English. We have called for Language service for help with English language editing.*

The second and more important issue relates to the statistical analyses of the change in density surrounding the burrows. While the authors provide R-squared values for their fits, they do not plot the fits, nor their confidence bounds. Instead the current figure simply has lines connecting the points, however the points are not in increasing order, so the lines bounce all around in a zigzag manner. What I am hoping to see is the regression line drawn through the data along with the confidence bounds. See https://www.mathworks.com/help/stats/polyconf.html for an example where both the fit line and confidence bounds are shown overlaid upon the data. The basic statistical question is not "does each slope have a robust fit", which is what the authors currently provide. Instead, the basic statistical question at hand is "are the two slopes significantly different"? This is at the heart of the manuscript – whether bioturbation has a statistically significant influence on density (and thus carbonate lithification). I don't think these should be prohibitively difficult for the authors to address, and I continue to hold the opinion that this work should be eventually suitable for publication in Biogeosciences. In my opinion we aren't there yet – but definitely getting closer.

*We are thankful for the valuable comments and suggestions for promoting the quality of data treatment. In the revised version of manuscript, confidence bounds are shown in the figure. With 95% confidence bonds, bioturbated area > 1000 (pixels unit) shows significantly*

*difference with unbioturbated area>1000. Integrated density extracted from the area <1000 seems not significant.*

*Integrated density obtained by ImageJ is the summation of calibrated gray values. No matter how large the diameter of each burrow is, we measured the gray values of the 10 pixels (~0.3 cm) around the burrows. Thus, compared to the burrows with small diameter, areas around burrows with bigger diameters are more representative. Nevertheless, the difference of the integrated density can be shown from Fig. 5.*

*In order to make the comparison more clear, an independent t-test was run on the data (Integrated density/Area ratios in Fig. 5) with a 95% confidence interval (CI) for the mean difference. The mean difference is 0.488-0.588=0.100. The p-value of Levene's test is 0.003 < 0.005, so we reject the null of levene's test and conclude that the variance is significantly different. The negative t value in the test indicates that the mean values for the first group, bioturbated, is significantly lower than the second group, control. The 95% CI is [-0.0124, -0.0753], which does not contain zero, this agree with the small p-value (0.000) of the significance test.*

| | *Data numbers* | *Mean* | *Std.Deviation* | *Std. Erro Mean* |
|---|---|---|---|---|
| *Bioturbated* | *113* | **0.488** | *0.054* | *0.005* |
| *Control* | *59* | **0.588** | *0.087* | *0.011* |

| | *Levene's test for equality of variance* | | *t-test for Equality of means* | | | | | | |
|---|---|---|---|---|---|---|---|---|---|
| | *F* | *Sig.* | *t* | *df* | *Sig.(2-tailed)* | *Mean difference* | *Std. Error Difference* | *95% confidence interval of the Difference* | |
| | | | | | | | | *Lower* | *Upper* |
| *Equal variances assumed* | *9.110* | *0.003* | *-9.285* | *170* | *0.000* | *-0.100* | *0.0108* | *-0.1213* | *-0.0788* |
| *Equal variances not assumed* | | | **-8.048** | **81.633** | **0.000** | **-0.100** | **0.0124** | **-0.1248** | **-0.0753** |

Specific comments

Pg. 1, Line 10: "blanketing the seafloor of the"

*Thanks for your correction.*

Pg. 1, Line 12: "in this carbonate lithified area"

*Thanks for your correction.*

Pg. 1, Line 13: "were examined" … also "enhances".

*Thanks for your correction.*

Pg. 2, Line 18: This is a bit awkward, I recommend "We examined this intriguing occurrence of nonburial carbonate… and highlight the interactions…”

*"In this research, we examined this intriguing occurrence of non-burial carbonate lithification in the deep-sea and highlight the interactions that take place between bioturbation and lithification on the mid-ocean ridge."*

Pg. 2, Line 28: "substantially in"
*Thanks for your correction.*

Pg. 3, Line 19: "which is a public"
*Thanks for your correction.*

Pg. 3, Line 27: "The MATAB function polyfit was used"
*Thanks for your correction.*

Title of section 5.2: "around burrows"
*YES, Maybe you refer to the section 4.2 and it has been changed.*

Pg. 5, lines 29–31: There are multiple grammatical problems in this sentence.
*It has been rephrased*
*"Therefore, the carbonate deposits on the SWIR could represent bioclastic deposition from "biogenic bloom", which was the productivity related event in a large part of Indian Ocean during the middle Miocene to the early Pliocene (Singh et al., 2012; Rai and Singh, 2001; Gupta et al., 2004; Arumugm et al., 2014). "*

Pg. 6, Line 4: "from SEM images"
*Thanks for your correction.*

Pg. 6, Line 20: "hydrothermal systems and detrital input"
*Thanks for your correction.*

Pg. 7, Line 24: "bulk samples are"
*Thanks for your correction.*

Pg. 7, Line 31: "deep-sea environments"
*Thanks for your correction.*

Pg. 8, Line 7: "of the studied carbonate area"
*Thanks for your correction.*

Pg. 8, Line 9: "phases", then line 10: "this is not likely to occur here"
*Thanks for your correction.*

Pg. 8, Line 12: "However, the carbonate samples studied here have never been buried"
*Thanks for your correction.*

Pg. 8, Line 15: "Moreover, ecological niches"

*Thanks for your correction.*

Pg. 8, Line 18: The beginning of this paragraph has grammatical problems.

*Thanks for your correction.*

Conclusion: Multiple grammatical issues.

*Thanks for your correction.*

Figure 5: There are lines connecting the points that show a zig-zag pattern as they trace the order of the points without any sorting. In other words, they are simply connecting the points in a random order. These lines should be removed and the actual fits presented (the trendlines that go through these point clouds, which are not currently shown).

*Thanks for reminding. Fig.5 has been edited following your suggestion.*

[revised manuscript text omitted]

---

## Author Response (AR3)

Dear editor:

We are very glad to receive your decision about our paper "Macrofaunal burrowing Enhances Deep-sea Carbonate Lithification on the Southwest Indian Ridge". Thank you very much for your hard work and timely reply. In revised version of manuscript:

1. Manuscript is improved following technical corrections from the editors. Figure 5 is also revised.

2. Figure 6, 7, and 8 are also revised. In the early version of manuscript, "Fe-Mn crust" was used in Fig.6-8. In revised version, "Mn- and Fe-oxide" is used following the suggestion from anonymous reviewer. So Figures are changed accordingly.

3. Figure 1 is modified after Zhu et al., (2010). The reference is added.

4. Because the first author Hengchao Xu is studied at Tongji University now, and some geochemical studies were done at Tongji University, so we want to add Tongji as the second institutional addresses.

5. We hope we can add a Foundation name. During this work, we has been helped by right of this foundation, but we forgot to write it. We also wonder if we can do it.

Thank you very much for your consideration.

Sincerely Yours

Xiaotong Peng, on behalf of the co-authors.

Editor:

I have a few minor suggestions for change that you may still consider (as technical corrections):

on page 2:

- line 2: "The explanations" could be changed to "These processes"

*The change is done accordingly.*

- line 2: "the fact that" could be changed to "the observation that"

*The change is done accordingly.*

- line 7: "debatable" could be changed to "debated"

*The change is done accordingly.*

- line 8: "enhanced" could be changed to "enhance"

*The change is done accordingly.*

on page 6:

- line 2: change to "from the "biogenic bloom"

*The change is done accordingly.*

Figure 5: please indicate in the caption what the different symbols in the figure indicate.

*Figure 5 is revised.*

[revised manuscript text omitted]